# Online Control for Meta-optimization

**Xinyi Chen**
Princeton University
Google DeepMind
xinyic@princeton.edu

**Elad Hazan**
Princeton University
Google DeepMind
ehazan@princeton.edu

## Abstract

Choosing the optimal hyperparameters, including the learning rate and momentum, for specific optimization instances is a significant yet nonconvex challenge. This makes conventional iterative techniques such as hypergradient descent insufficient in obtaining global optimality guarantees in general.

We consider the more general task of meta-optimization – online learning of the best optimization algorithm given problem instances. For this task, a novel approach based on control theory is introduced. We show how meta-optimization can be formulated as an optimal control problem, departing from existing literature that use stability-based methods to study optimization. Our approach leverages convex relaxation techniques in the recently-proposed nonstochastic control framework to overcome the challenge of nonconvexity, and obtains regret guarantees vs. the best offline solution. This guarantees that in meta-optimization, we can learn a method that attains convergence comparable to that of the best optimization method in hindsight from a class of methods.

## 1 Introduction

The performance of optimization methods, in particular for the application of training deep neural networks, crucially depends on their hyperparameters. Bengio [2012] notes that the learning rate is the "single most important parameter" for efficient training. However, this parameter is also notoriously hard to tune without intensive hyperparameter search. An even more difficult problem is adapting all aspects of an optimization algorithm, including momentum, preconditioner, weight decay, and more, to potentially changing tasks. This is the motivating problem of our paper.

The problem of learning the best optimizer, even for the special case of hyperparameter tuning of the learning rate, is difficult because **optimizing these hyperparameters is nonconvex**. We tackle this challenging problem by a novel framework that is based on several components:

- We formalize the problem of "finding the best algorithm" as the task of **meta-optimization**. In this task, the player is given a sequence of optimization problems. Her goal is to solve them as fast as the best algorithm in hindsight from a family of possible methods. This setting generalizes hyperparameter tuning to encompass several attributes, including the preconditioning matrix and momentum.

- We propose a new metric for meta optimization called **meta-regret**, which measures the total cost compared to the cost of the best algorithm in hindsight in meta-optimization. We give an efficient method for minimizing the meta-regret that attains sublinear guarantee, which is tight. The implication of this efficient method with optimal meta-regret bound is that we can compete with the best hyperparameters in hindsight (although not necessarily find them).

- We give a **novel control formulation** for meta-optimization based on the recently proposed framework of online nonstochastic control. This control formulation differs in several

aspects to previous control methods applied to optimization: it is based on optimal control (rather than Lyapunov stability), and requires robustness to nonstochastic perturbations.

- The online control formulation gives rise to our **efficient method** which is based on the Gradient Perturbation Controller (GPC), a new type of online feedback-loop controller that requires only gradient access from the objective functions.

  While the method itself is based on recent advancements in control, its analysis requires **new technical components**. Meta-optimization requires learning between optimization episodes, which necessitates the analysis of feedback control in changing environments without bounded disturbances. This presents a challenge of bounding the magnitude of the state, which we address for the first time in the online control literature.

## 1.1 The setting of meta-optimization

In meta-optimization, we are given **a sequence of optimization problems**, called episodes. The goal of the player is not only to minimize the total cost in each episode, but also to compete with a set of available optimization methods.

Specifically, in each of the $N$ episodes, we have a sequence of $T$ optimization steps that are either deterministic, stochastic, or online. Throughout the paper, we use $(t, i)$ to denote time $t$ in episode $i$. At the beginning of an episode, the iterate is "reset" to a given starting point $x_{1,i}$. In the most general formulation of the problem, at time $(t, i)$, an optimization algorithm $\mathcal{A}$ chooses a point $x_{t,i} \in \mathcal{K}$, in a convex domain $\mathcal{K} \subseteq \mathbb{R}^d$. It then suffers a cost $f_{t,i}(x_{t,i})$. Let $x_{t,i}(\mathcal{A})$ denote the point chosen by $\mathcal{A}$ at time $(t, i)$, the protocol of this setting is formally defined in Algorithm 1.

---
**Algorithm 1** Meta-optimization
---
1: Input: $N, T, \mathcal{A}, \mathcal{K}$, reset points $\{x_{1,i}\}_{i=1}^{N}$.
2: **for** $i = 1, \ldots, N$ **do**
3:     **for** $t = 1, \ldots, T$ **do**
4:         Play $x_{t,i} = \mathcal{A}(f_{1,1}, \ldots, f_{t-1,i}) \in \mathcal{K}$ if $t > 1$; else play $x_{1,i}$.
5:         Receive $f_{t,i}$, pay $f_{t,i}(x_{t,i})$.
6:     **end for**
7: **end for**
---

The standard goal in optimization, either deterministic, stochastic, or online, is to minimize the cost of each episode in isolation. In meta-optimization, the goal is to minimize the cumulative cost in both each episode, and overall in terms of the choice of the algorithm. We thus define meta-regret to be

$$\text{MetaRegret}(\mathcal{A}) = \sum_{i=1}^{N} \sum_{t=1}^{T} f_{t,i}(x_{t,i}(\mathcal{A})) - \min_{\mathcal{A}^* \in \Pi} \sum_{i=1}^{N} f_{t,i}(x_{t,i}(\mathcal{A}^*)),$$

where $\Pi$ is the benchmark algorithm class. The meta-regret is the regret incurred for learning the best optimization algorithm in hindsight, and captures both efficient per-episode optimization as well as competing with the best algorithm in $\Pi$.

Meta-regret can be studied under various optimization settings. In the deterministic setting, the objectives are the same across time steps and episodes; in the stochastic setting, the objectives are drawn from distributions that can vary at each $t$; finally, in the adversarial setting, the objectives appear online. Meta-regret guarantees can be refined for each setting; for example, in the deterministic setting, meta-regret translates to a bound on the cost of the average iterate.

**Why is meta-optimization hard?** It is natural to apply standard techniques from optimization, such as local search or gradient based methods, to meta-optimization. Indeed, this has been studied in numerous previous works under related settings, e.g. Baydin et al. [2017], Chandra et al. [2019].

However, the resulting optimization problem is nonconvex and local gradient optimization may reach suboptimal solutions. For example, a natural application of meta-optimization is to learn hyperparamters of a class of optimization algorithms. In particular, finding the optimal gradient descent learning rate for a specific objective takes the form of the following minimization problem:

$$\min_{\eta} f(x_T), \quad \text{subject to } x_{t+1} = x_t - \eta \nabla f(x_t).$$

We can unroll the minimization objective from the initial point $x_1$,

$$f(x_T) = f(x_1 - \eta \nabla f(x_1) - \eta \sum_{t=2}^{T-1} \nabla f(x_t)) \quad = f(x_1 - \eta \nabla f(x_1) - \eta \nabla f(x_1 - \eta \nabla f(x_1)) - ...)$$

From this expression, it is clear that $f(x_T)$ is a nonconvex function of $\eta$.

The setting of meta-optimization relates to and/or generalizes previous approaches for hyperparamter tuning in optimization. For example, it is the online learning analogue of the average case analysis framework of optimization algorithms [Pedregosa and Scieur, 2020, Paquette et al., 2021], and generalizes control-based approaches for analyzing a single optimization instance [Lessard et al., 2016, Casgrain and Kratsios, 2021].

## 1.2 Main theorem statements

Applying iterative optimization methods directly to the meta-optimization problem is unlikely to result in global guarantees, since the problem is nonconvex. We take a different approach: we formulate meta-optimization as an online optimal control problem. Since the control formulation has disturbances that are not stochastic, as shown in Section 2, we adopt the recently-developed nonstochastic control framework, and use techniques therein to develop efficient methods for unconstrained meta-optimization of quadratic and convex smooth functions.

The following are informal statements of our results, where the algorithm referenced is detailed in Section 3. Due to space constraints, we include the bandit algorithm in Appendix D.

---

**Theorem 1.1** (Informal). *For quadratic losses, Algorithm 2 satisfies*

$$MetaRegret(\mathcal{A}) = \sum_{i=1}^{N} \sum_{t=1}^{T} f_{t,i}(x_{t,i}^{\mathcal{A}}) - \min_{\mathcal{A}^* \in \Pi} \sum_{i=1}^{N} \sum_{t=1}^{T} f_{t,i}(x_{t,i}^{\mathcal{A}^*}) = \tilde{O}(\sqrt{NT}).$$

**Theorem 1.2** (Informal). *For convex smooth losses, a bandit variant of Algorithm 2 satisfies*

$$\mathbb{E}\left[MetaRegret(\mathcal{A})\right] = \tilde{O}((NT)^{3/4}).$$

---

In both statements, $\tilde{O}$ hides factors polynomial in parameters of the problem and logarithmic in $T, N$. The sublinear scaling in both $N$ and $T$ implies that with more episodes, the average performance approaches that of the best optimizer from a family of optimization methods in hindsight. Our theorem holds for the most general optimization setting, the adversarial setting where $f_{t,i}$ appears in an online fashion. The theorem can be refined for the deterministic and stochastic settings, for more details see Appendix F.

## 1.3 Related Work

**Online convex optimization and nonstochastic control.** Our methods for meta-optimization are based on iterative gradient (or higher order) methods with provable regret guarantees. These have been developed in the context of the online convex optimization framework. In this framework, a decision maker iteratively chooses a point $x_t \in \mathcal{K}$, for a convex set in Euclidean space $\mathcal{K} \subseteq \mathbb{R}^d$, and receives loss according to a convex loss function $f_t : \mathcal{K} \mapsto \mathbb{R}$. The goal of the decision maker is to minimize her regret, defined as

$$\text{Regret} = \sum_{t=1}^{T} f_t(x_t) - \min_{x \in \mathcal{K}} \sum_{t=1}^{T} f_t(x).$$

Notably, the best point $x^* \in \mathcal{K}$ is defined only in hindsight, since the $f_t$'s are unknown a priori. For more information on this setting as well as an algorithmic treatment see [Hazan et al., 2016].

Techniques from online convex optimization were instrumental in developing an online control theory that permits nonstochastic disturbances and is regret-based. Deviating from classical control theory,

the online nonstochastic control framework treats control as an interactive optimization problem, where the objective is to make decisions that compete with the best controller in hindsight. For a detailed exposition see [Hazan and Singh, 2022]. In general, computing the best controller in hindsight is a nonconvex optimization problem if we allow general convex costs. However, online nonstochastic control has provable guarantees in this setting despite the nonconvexity, since it uses policies in a larger policy class via convex relaxation. This is crucial for our guarantees: meta-optimization can be a nonconvex problem, but we can hope to use feedback control to bypass the nonconvexity and obtain meta-regret bounds.

**Average case analysis of optimization.** A closely related framework for analyzing optimization methods is the average-case analysis framework developed in [Pedregosa and Scieur, 2020]. This framework studies the expected performance of optimization algorithms when the problem is sampled from a distribution, and allows for more fine-grained results than typical worst-case analysis. Average-case optimal first-order methods for minimizing quadratic objectives are proposed in [Pedregosa and Scieur, 2020], and [Domingo-Enrich et al., 2021] extends the study to bilinear games. The meta-optimization setting is relevant to, but significantly more general than the average-case analysis framework, since we do not assume known stochastic distribution of the optimization problems, and we compete with the best algorithm in hindsight.

**Hypergradient descent and hyperparameter tuning for optimizer parameters.** Hyperparameter optimization is a significant problem in deep learning practice, and thus was intensively studied. The work of Baydin et al. [2017] apply local iterative methods to the problem of optimizing the learning rate from an empirical standpoint, and [Chandra et al., 2019] gives better practical methods. However, even the simple case of optimizing the learning rate can be nonconvex, and it is unlikely that iterative methods will yield global optimality guarantees. Certain provable guarantees for quadratic functions and scalar learning rate are presented in [Wang et al., 2021b].

**Control for optimization.** The connections between control and optimization go back to Lyapunov's work and its application to the design and analysis of optimization algoriths. We survey the various approaches in detail in Appendices A and B. Lessard et al. [2016] apply control theory to the analysis of optimization algorithms on a single problem instance. They give a general framework, using the notion of Integral Quadratic Constraints from robust control theory, for obtaining convergence guarantees for a variety of gradient-based methods. Casgrain and Kratsios [2021] study the characterization of the regret-optimal algorithm given an objective function, using a value function-based approach motivated by optimal control. In contrast, we aim to learn the optimal algorithm from a sequence of optimization problems, with potentially different objectives and characteristics.

**Learning to optimize.** Learning to Optimize (L2O) is a related framework with a focus on empirical methods. The goal is to learn an optimizer, usually parameterized by a deep neural network, automatically on a set of training problems that can also generalize to unseen problem instances. Andrychowicz et al. [2016] propose learning updates in a coordinate-wise fashion using an LSTM architecture. Wichrowska et al. [2017] later suggest several modifications to improve the generalization capabilities of learned optimizers. More recently, Metz et al. [2022] proposes a learned optimizer trained on a wide variety of tasks, using evolutionary strategies for estimating gradient updates to the meta-optimizer.

**Meta-learning.** Meta-learning Thrun and Pratt [1998], Finn et al. [2017] aims to learn algorithms or model parameters that can adapt rapidly to new tasks. Regret guarantees over model parameters for online meta-learning were developed in Finn et al. [2019], and guarantees for adaptive gradient-based meta-learning were given in Balcan et al. [2019]. The latter work provides guarantees for meta-learning optimization algorithms, but the performance metric in this context is an upper bound on the actual performance. In contrast, meta-regret measures the performance gap directly and minimizing it is a nonconvex problem, necessitating the use of techniques from online nonstochastic control theory.

## 1.4 Structure of the paper

In Section 2, we introduce the new control formulation of meta-optimization, and elaborate on the assumptions needed to apply the nonstochastic control framework. In Section 3, we state the algorithm and main results, as well as the benchmark algorithm class. Illustrative experimental

results of our meta-optimization methods are given in Section 4. In Appendix A and B, we give more background on the historical use of control theory in optimization. We describe the recent framework of online nonstochastic control in Appendix C, and why it is important for meta-optimization. Generalizations of our dynamical system formulation to capture smooth convex meta-optimization is given in section D. More expressive optimizers are given in Appendic E. The extension to the stochastic and deterministic settings is given in F. Finally, we include more experiments, their descriptions and settings in Appendix J.

## 2 Online control formulation of meta-optimization

In this section, we explain our control formulation of the meta-optimization problem, representing a first step to developing an efficient control-based method. Different from historical perspectives, which we outline in Appendices A and B, our formulation gives rise to an optimal control problem rather than a stability analysis.

In its simplest form, an optimal control problem is a minimization problem over the controls:

$$\min_{u_1,\dots,u_T} \quad \sum_{t=1}^{T} c_t(z_t, u_t) \tag{1}$$
$$\text{s.t.} \quad z_{t+1} = Az_t + Bu_t + w_t.$$

The states are denoted as $z_t$, and for a linear dynamical system they evolve according to the dynamics determined by $A$ and $B$. The control signals and disturbances are denoted as $u_t$ and $w_t$, respectively. The control objective is the sum of control costs $c_t$, which are functions of the states and controls. This optimization problem has explicit solutions under certain conditions, if we know the cost functions a-priori; however, since meta-optimization is inherently online, we turn to the more recent regret-based online control setting.

In online control, the convex cost functions can appear in an adversarial fashion, and the controller can choose the controls iteratively as the cost functions are revealed. The goal in online control is to minimize policy regret:

$$\sum_{t=1}^{T} c_t(z_t, u_t) - \min_{\pi \in \Pi_C} \sum_{t=1}^{T} c_t(z_t^\pi, u_t^\pi), \tag{2}$$

where $\Pi_C$ denotes the comparator policy class, $z_t^\pi, u_t^\pi$ denote the state and controls under the policy $\pi$, respectively. The policy regret is the excess total cost of the control algorithm, compared to that of the best controller in $\Pi_C$ in hindsight. In the following discussion, we specify how meta-optimization can be formulated as an online control problem.

For simplicity, consider the online convex optimization setting where all objective functions $f_{t,i} : \mathbb{R}^d \to \mathbb{R}$ are convex quadratic. Denote the location-independent Hessian as $\nabla^2 f_{t,i}(x) = H_{t,i} \in \mathbb{R}^{d \times d}$. This setting can be generalized to convex smooth functions, as we detail in Appendix D. We assume that the functions have bounded Hessian, and the gradients satisfy:

**Assumption 1.** $\forall i \in [N]$, $t \in [T]$, $x \in \mathbb{R}^d$, $\|\nabla f_{t,i}(x)\| \le 2\beta\|x\| + b$.

**The dynamical system.** The state of the dynamical system at time $(t, i)$ is given by the vector below consisting of the current optimization iterate, the previous iterate, and the previous gradient,

$$z_{t,i} = \begin{bmatrix} x_{t,i} \\ x_{t-1,i} \\ \nabla f(x_{t-1,i}) \end{bmatrix}.$$

For an extension of the control formulation to include more iterates and gradients, see Appendix E.

Let $\delta > 0$ be a regularization parameter, and $\eta > 0$ be the base learning rate, consider the system:

$$\begin{bmatrix} x_{t+1,i} \\ x_{t,i} \\ \nabla f_{t,i}(x_{t,i}) \end{bmatrix} = \begin{bmatrix} (1-\delta)I & 0 & -\eta I \\ I & 0 & 0 \\ H_{t,i} & -H_{t,i} & 0 \end{bmatrix} \times \begin{bmatrix} x_{t,i} \\ x_{t-1,i} \\ \nabla f_{t-1,i}(x_{t-1,i}) \end{bmatrix} + \begin{bmatrix} I & 0 & 0 \\ 0 & 0 & 0 \\ 0 & 0 & 0 \end{bmatrix} \times u_{t,i} + \begin{bmatrix} 0 \\ 0 \\ \nabla f_{t,i}(x_{t-1,i}) \end{bmatrix},$$
$$\tag{3}$$

which is a valid representations of the optimization process given the relation

$$\nabla f_{t,i}(x_{t,i}) = \nabla f_{t,i}(x_{t-1,i}) + H_{t,i}(x_{t,i} - x_{t-1,i}).$$

The disturbance in the system evolution includes $\nabla f(x_{t-1,i})$ as a component, which is not stochastic in nature. Consequently, classical control approaches such as $\mathcal{H}_2$ optimal control are not applicable, and we adopt the online nonstochastic control framework [1]. This formulation also enables us to capture the state reset when a new episode begins by using adversarial disturbances.

Without the control signal $u_{t,i}$, the system describes the partially time-delayed gradient descent update with $\ell_2$ regularization: $x_{t+1,i} = (1 - \delta)x_{t,i} - \eta \nabla f(x_{t-1,i})$. Several results show that under mild conditions, the regret using delayed gradients are constant factors away from online gradient descent using using fresh gradients [Quanrud and Khashabi, 2015, Langford et al., 2009]. The parameter $\delta$ is user-specified to ensure stability, and can be arbitrarily small.

The control signal $u_{t,i}$ contributes only to the update of $x_{t,i}$, and does not affect the rest of the state. Observe that we can simulate any optimizer's update by using the control signal, and therefore we can learn the best optimizer in hindsight by optimizing over the $u_{t,i}$'s.

In meta-optimization, the dynamical system evolves according to (3) during an episode. However, each episode starts with the optimization iterate at an arbitrary initialization $x_{1,i}$, and we need to reset the system before a new episode begins. To transition the system state to the new initialization, consider the following reset disturbance,

$$w_{T,i} = \begin{bmatrix} x_{1,i+1} - ((1 - \delta)x_{T,i} - \eta \nabla f_{T-1,i}(x_{T-1,i}) + \bar{u}_{T,i}) \\ x_{1,i+1} - x_{T,i} \\ \nabla f_{T,i}(x_{T-1,i}) - \nabla f_{T,i}(x_{T,i}) \end{bmatrix}, \tag{4}$$

where $\bar{u}_{T,i}$ is the top $d$ entries of the control signal $u_{T,i}$. Under this reset disturbance, the initial state of an episode is $z_{1,i} = \begin{bmatrix} x_{1,i}^\top & x_{1,i}^\top & 0 \end{bmatrix}^\top$, which is consistent with the meta-optimization protocol. Finally, we assume that the initial point in each episode has bounded norm.

**Assumption 2.** *For all $i$, $x_{1,i}$ satisfies $\|x_{1,i}\| \leq R$.*

**Online control formulation.** Consider the online control problem (2), where the system evolves according to (3) and the control costs are the optimization objectives,

$$\sum_{i=1}^{N} \sum_{t=1}^{T} f_{t,i}(x_{t,i}) - \min_{\pi \in \Pi_C} \sum_{i=1}^{N} \sum_{t=1}^{T} f_{t,i}(x_{t,i}^\pi). \tag{5}$$

Note that the cost functions are functions of the states: $c_{t,i}(z_{t,i}, u_{t,i}) = f_{t,i}(Sz_{t,i}) = f_{t,i}(x_{t,i})$, where $S$ is a matrix that selects the first $d$ entries of $z_{t,i}$. In addition, observe that even though the cost functions do not explicitly depend on the controls $u_{t,i}$, they still affect the states through the control policy. This online control problem directly corresponds to meta-regret minimization, where the benchmark algorithm class $\Pi$ consists of optimizers whose updates match the dynamics of system (3) under controllers in $\Pi_C$.

**Stability.** In nonstochastic control, a stable system or access to a stabilizing controller is necessary to achieve regret that is polynomial in the state dimension Chen and Hazan [2021]. The notion of stability in this context is stronger than the conventional notion, and for linear time-varying (LTV) dynamical systems, it is referred to as sequential stability. We assume our system satisfies this condition.

**Definition 2.1** (Sequentially stable). *An LTV system whose dynamics at time $t$ is $z_{t+1} = A_t z_t + B_t u_t + w_t$ is $(\kappa, \gamma)$ sequentially stable if for all intervals $I = [r, s] \subseteq [T]$, $\|\prod_{t=s}^{r} A_t\| \leq \kappa^2 (1-\gamma)^{|I|}$.*

**Assumption 3.** *For settings where the functions $f_{t,i}$ are changing, we assume that the resulting dynamical system is sequentially stable with $\kappa \geq 1$.*

---

[1] Notice that the regret bounds we prove are under the same sequence of disturbances, and apply to the gradients along the taken trajectory. This is similar to the nature of adaptive gradient methods: the best regularization in hindsight depends on the observed gradients, rather than the ones that would appear had we used a different algorithm.

The above assumption is standard in the literature of nonstochastic control for time-varying systems Gradu et al. [2020b], Minasyan et al. [2021]. If the dynamical system is linear time-invariant (LTI), then Assumption 3 is satisfied if the system is strongly stable, whose definition is given below.

**Definition 2.2** (Strong stability). *A system $z_{t+1} = Az_t + Bu_t + w_t$ is $(\kappa, \gamma)$ strongly stable if there exist matrices $P, Q$, such that $A = PQP^{-1}$, and $\|Q\| \leq 1 - \gamma$, $\|P\|, \|P^{-1}\| \leq \kappa$.*

In the deterministic meta-optimization setting, we have an LTI system. We show in the appendix that with $\eta$ scaling inversely as the smoothness of the function, and $\delta$ arbitrarily small, the LTI system is strongly stable for some $(\kappa, \gamma)$. The condition on $\eta$ is natural, since gradient descent with learning rate larger then $\frac{1}{\beta}$ diverges.

# 3 Algorithm and main theorem statements

In this section, we give an efficient algorithm that minimizes meta-regret and state accompanying guarantees. Our main algorithm for quadratic meta-optimization is given in Algorithm 2. This algorithm views meta-optimization as a single-trajectory control problem, and uses the Gradient Perturbation Controller (GPC) method adapted from Agarwal et al. [2019]. The GPC method considers a specific class of controllers, namely disturbance-feedback controllers, that are linear in the past disturbances, as defined below.

**Definition 3.1** (Disturbance-feedback controller). *A disturbance-feedback controller (DFC), $\pi(M)$ is parameterized by a sequence of $L$ matrices $\{M^l\}_{l=1}^{L}$, where $M^l$ denotes the l-th matrix, instead of a matrix power. It outputs the control signal $u_{t,i} = \sum_{i=1}^{L} M^l w_{t-l,i}$.*

The GPC method uses the class of DFC controllers, and makes gradient updates on the parameters of interest, $M$, to compete with the best DFC in hindsight.

Similar to the GPC method, at each time step, Algorithm 2 outputs the control signal generated by $\pi(M_{t,i})$, and receives an objective function. Then, the disturbances are computed according to the dynamical system formulation (3), and a cost function for updating $M$ is constructed on Line 9. Finally, a gradient update is executed on $M$.

The surrogate cost on Line 9 has the following expression:

$$g_{t,i}(M_{t-L,i}, \ldots, M_{t-1,i}) = c_{t,i}(y_{t,i}(M_{t-L,i}, \ldots, M_{t-1,i})), \tag{6}$$

where the surrogate state $y_{t,i}$ is the state reached at time $(t, i)$, if $z_{t-L,i} = 0$ and we execute the sequence of policies $M_{t-L,i}, \ldots, M_{t-1,i}$. Note that the function $g_{t,i}(M) = g_{t,i}(M, \ldots, M)$ is a function of only $M$, instead of a sequence of time-varying controls. As a result, when we take a gradient of $g_{t,i}(M)$, the gradient captures the effects of changing $M$ over multiple time steps.

For smooth convex optimization, we cannot directly observe the dynamics $A_{t,i}$. There are two consequences for our algorithm: we cannot access the full gradient with respect to $M$, and we cannot compute $y_{t,i}$ directly. However, a bandit variant of Algorithm 2 overcomes both difficulties, as the algorithm approximates $y_{t,i}(M, \ldots, M)$, and estimates the gradient of $M$ with zeroth-order methods. The algorithm is given in Appendix D.

## 3.1 Guarantees for quadratic and smooth meta-optimization

We first state the regret guarantees of our algorithms, and describe the benchmark algorithm class in more detail in the next subsection. All proofs are in the full version at https://arxiv.org/abs/2301.07902.

**Theorem 3.2** (Quadratic). *Under Assumptions 1, 2, 3, Algorithm 2 with $\eta \leq 1, \delta \in (0, \frac{1}{2}]$, $\eta_g = \Theta(\sqrt{NT})^{-1}$, and $L = \Theta(\log NT)$ satisfies*

$$MetaRegret = \sum_{i=1}^{N}\sum_{t=1}^{T} f_{t,i}(x_{t,i}) - \min_{\mathcal{A} \in \Pi} \sum_{i=1}^{N}\sum_{t=1}^{T} f_{t,i}(x_{t,i}^{\mathcal{A}}) \leq \tilde{O}(\sqrt{NT}),$$

*where $\Theta$ contains polynomial factors in $\gamma^{-1}, \beta, \kappa, R, b, d$, and $\tilde{O}$ contains these polynomial factors as well as logarithmic factors in $T, N$.*

---

**Algorithm 2** Gradient perturbation controller for meta-optimization

---
1: **Input:** $N, T, z_{1,1}, \eta, \delta, \eta_g$, starting points $\{x_{1,i}\}_{i=1}^{N}, \kappa, \gamma$
2: **Set:** $\mathcal{M} = \{M = \{M^1, \ldots, M^L\} : \|M^l\| \leq \kappa^3(1-\gamma)^l\}$, and initialize any $M_{1,1} \in \mathcal{M}$.
3: **for** $i = 1, \ldots, N$ **do**
4:     If $i > 1$, set $z_{1,i} = z_{T+1,i-1}$, $M_{1,i} = M_{T+1,i-1}$.
5:     **for** $t = 1, \ldots, T$ **do**
6:         Choose $u_{t,i} = \sum_{l=1}^{L} M_{t,i}^l w_{t-l,i}$ and observe $z_{t+1,i}$.
7:         Receive $f_{t,i}$, compute $\nabla f_{t,i}(x_{t,i}), \nabla f_{t,i}(x_{t-1,i})$. If $t = T$, compute $w_{T,i}$ according to (4).
8:         Suffer control cost $c_{t,i}(z_{t,i}) = f_{t,i}(x_{t,i})$.
9:         Construct ideal cost $g_{t,i}(M) = g_{t,i}(M, \ldots, M)$ according to (6).
10:        Perform gradient update on the controller parameters, where $\Pi$ denotes projection :

$$M_{t+1,i} = \Pi_{\mathcal{M}}(M_{t,i} - \eta_g \nabla g_{t,i}(M_{t,i})).$$

11:     **end for**
12: **end for**

---

It follows from lower bounds in online convex optimization Hazan et al. [2016] that the dependence of meta-regret on $N, T$ are optimal in Theorem 3.2. The next theorem holds for meta-optimization over convex smooth objectives, and the additional assumptions are given in Appendix D. Algorithm D.2 and the proof of Theorem 3.3 are based on nonstochastic control techniques in the bandit setting Gradu et al. [2020a].

**Theorem 3.3** (Smooth). *Under Assumptions 2, 3, 4, 5, Algorithm D.2 with $\eta \leq 1$, $L = \Theta(\log NT)$, there exists a setting of $\delta_M, \{\eta_{t,i}^g\}$ that satisfies*

$$\mathbb{E}\left[MetaRegret\right] = \mathbb{E}\left[\sum_{i=1}^{N}\sum_{t=1}^{T} f_{t,i}(x_{t,i})\right] - \min_{\mathcal{A} \in \Pi}\sum_{i=1}^{N}\sum_{t=1}^{T} f_{t,i}(x_{t,i}^{\mathcal{A}}) \leq \tilde{O}((NT)^{3/4}),$$

*where $\tilde{O}$ contains polynomial factors in $\gamma^{-1}, \beta, \kappa, R, b, d$, and logarithmic factors in $T, N$.*

**Remark 3.4.** *Stronger performance metrics that are more suitable for changing environments, such as adaptive regret and dynamic regret, were explored in the context of control of time varying dynamical systems, see [Gradu et al., 2020b, Minasyan et al., 2021]. These latter results can potentially be adapted to give stronger performance guarantees in meta-optimization.*

**Analysis overview and technical novelty.** The regret of Algorithm 2 is the excess total cost of our controller compared to that of the best fixed DFC in hindsight. Since the system is strongly stable, the control cost at $(t, i)$ depends minimally on the controls executed before $L$ time steps, e.g. $M_{t-L,i}, M_{t-L-1,i} \cdots$. Therefore, we can focus on a surrogate cost that is a function of only the past $L$ control signals, where $L$ is logarithmic in the horizon $NT$. Bounding the regret then reduces to obtaining low regret on such surrogate costs, which was studied in the online learning with memory (OCOwM) framework Anava et al. [2015].

Crucially, this analysis relies on universal upper bounds on the states and disturbances, which is the main technical challenge of this work. Indeed, this assumption does not hold in our setting, since the disturbances include $\nabla f_{t,i}(x_{t-1,i})$. Under the smoothness assumption, $\|w_{t,i}\|$ can grow proportionally with $\|x_{t-1,i}\|$. Furthermore, the reset disturbance can be on the same scale as the states themselves.

We overcome this challenge by showing that due to system stability, the effect of the reset disturbance attenuates exponentially. In each episode, after the initial large reset disturbance, the state and disturbance have an upper bound that decreases with time and eventually converges to a constant value. Moreover, if we scale the objective functions properly, we can bound the contribution from $w_{t,i}$ to the size of the state because of smoothness. To make this intuition formal, we use several inductive arguments to address the interdependence between the states and disturbances.

## 3.2  The benchmark algorithm class

We introduce the benchmark algorithm class we compete against, which consist of optimizers that correspond to a class of DFCs. Informally, our guarantee is competitive with optimizers that are linear functions of past gradients.

The comparator policy class we consider for control is $\Pi_{DFC}$, the set of all disturbance-feedback policies defined by $M$ such that $M \in \mathcal{M}$ defined in Algorithm 2. The GPC method obtains sublinear regret against $\Pi_{DFC}$ under a fixed sequence of disturbances, cost functions, and dynamics. However, if these sequences are fixed in meta-optimization, the gradient component in the state is no longer the true gradient; instead, they are *pseudo-gradients*. Let $x_{t,i}^M$ denote the optimization iterate at time $t$ in epoch $i$ under controller $\pi(M) \in \Pi_{DFC}$. Define the pseudo-gradient at time $(t, i)$ as

$$\hat{\nabla} f_{t,i}^M(\cdot) = \nabla f_{t,i}(\cdot) - \nabla f_{t,i}(x_{t-1,i}^M) + \nabla f_{t,i}(x_{t-1,i}),$$

where $x_{t-1,i}$ is the optimization iterate of our algorithm. Then the states under $\pi(M)$ are

$$z_{1,i}^M = \begin{bmatrix} x_{1,i} \\ x_{1,i} \\ 0 \end{bmatrix}, \quad z_{2,i}^M = \begin{bmatrix} x_{2,i}^M \\ x_{1,i} \\ \nabla f_{1,i}(x_{1,i}) \end{bmatrix}, \quad z_{t,i}^M = \begin{bmatrix} x_{t,i}^M \\ x_{t-1,i}^M \\ \hat{\nabla} f_{t-1,i}^M(x_{t-1,i}^M) \end{bmatrix} \text{ for } t \geq 3.$$

Observe that each policy in $\Pi_{DFC}$ corresponds to an optimization algorithm. Writing out the control signals, the benchmark algorithm class consists of optimizaers with updates:

$$x_{t+1,i}^M = (1 - \delta) x_{t,i}^M - \eta \hat{\nabla} f_{t-1,i}^M(x_{t-1,i}^M) + \sum_{l=1}^{L} M^l w_{t-l,i}.$$

This benchmark algorithm class has a more straightforward interpretation for deterministic meta-optimization. It can capture common algorithms on time-delayed pseudo-gradients, including gradient descent, momentum, and preconditioned methods. For details on the range of hyperparameters it contains, see Appendix G; for a concrete example of learning the learning rate, see Appendix H.

# 4  Experiments

We evaluate the empirical performance of our meta-optimization algorithm on a suite of tasks, including regression and classification. For an illustrative example, we consider regression with the objective $\frac{1}{2}\|Ax - b\|^2$. We first generate $A$ and $b$ element-wise using the standard Gaussian distribution. To simulate the stochastic meta-optimization setting, we have $N$ episodes and $T$ time steps, and the optimization iterate is re-initialized to the same point at the beginning of each episode. The initial value is randomly sampled. For each $t \in [T]$, and $i \in [N]$, we add entry-wise standard Gaussian noise to $b$. We set $d = 100, T = 500, N = 10$, and compare with gradient descent, momentum, Marchenko-Pastur, and nesterov accelerated gradient. The Marchenko-Pastur (MP) algorithm is optimal in the average-case sense for regression tasks whose design matrix has entries sampled from an i.i.d Gaussian distribution, and therefore should perform well in our setting without noise Pedregosa and Scieur [2020]. We tune the learning rate and momentum hyperparameters for all algorithms, and the sweep details are in Appendix J. For meta-optimization, we use $L = 3, \delta = 0$, and a base learning rate of $0.001$. We plot the moving average of the objective values in Figure 7, with a window size of 50.

The plot shows that initially, meta-optimization has similar performance to other algorithms. However, with an increasing number of episodes, the algorithm gradually learns and outperforms baselines both at the beginning and, eventually, towards the end of each episode. This illustrative experiment shows that for this particular problem, meta-optimization can improve and learn updates that minimize the objectives quickly.

For a proof-of-concept nonconvex task, we consider MNIST classification with a neural network. The network has 3 layers and is fully connected, with layer sizes [784, 512, 10]. We run for 5 episodes with 20 epochs in each episode. Since this problem is high-dimensional, we choose to learn scalar weights of past gradients. We take $L = 20$, and the network is trained using stochastic gradients from batches of size 256. The hyperparameters of the baselines are tuned, and the sweep details are in Appendix J. We also include Adam in our baseline, since it is a popular optimizer.

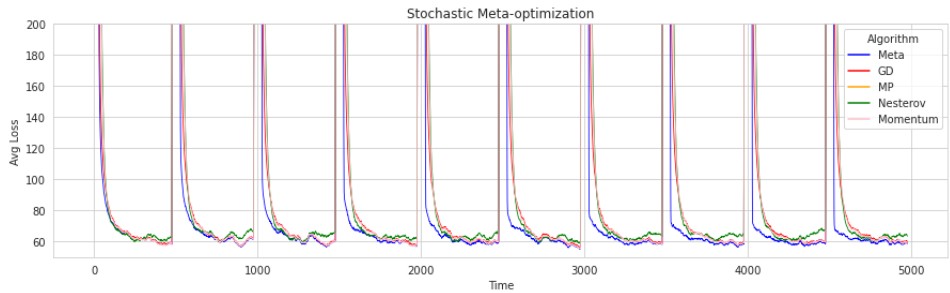

Figure 1: Comparison plot for the performance of meta-optimization and other baselines.

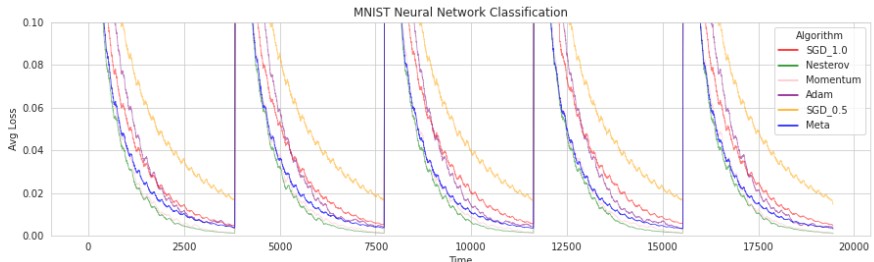

Figure 2: Comparison plot for the training performance of meta-optimization and other baselines with a suboptimal base learning rate.

In this experiment, the base learning rate $\eta$ for the meta optimizer is set to 0.5 instead of the optimal learning for SGD, which is 1.0. We show the training curves of SGD with learning rates of both 1.0 and 0.5 for comparison. Examining the plot, it is clear that the meta optimizer outperforms SGD with learning rate 0.5 by a wide margin, and continues to converge to the performance of momentum over episodes. We include more experiments in Appendix J.

## 5    Conclusion.

This manuscript proposes a framework for optimization whose goal is to learn the best optimization algorithm from experience, and gives an algorithmic methodology using feedback control for this meta-optimization problem. In contrast to well-studied connections between optimization and Lyapunov stability, our approach builds upon new techniques from the regret-based online nonstochastic control framework. We derive new efficient algorithms for meta-optimization using recently proposed control methods, and prove regret bounds for them.

An interesting direction for future investigation is extending the work on nonstochastic control of time varying and nonlinear dynamical systems to our setting of meta-optimization. In particular, it is interesting to explore the implications of low adaptive and dynamic regret in meta-optimization. Another area of investigation is the connection between meta-optimization using control, and adaptive methods: do the solutions proposed by the control approach have connections to adaptive regularization-based algorithms? Many intriguing problems arise in this new intersection of optimization and nonstochastic control.

## Acknowledgments and Disclosure of Funding

Elad Hazan acknowledges funding from the ONR award N000142312156, the NSF award 2134040, and Open Philanthropy.

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
