## A   Mathematical optimization and feedback control

The fields of mathematical optimization and control theory are closely related. Many optimization methods are inspired by natural dynamical systems, such as Polyak's heavy ball method [Polyak, 1964]. On the other hand, the analysis of optimization algorithms also has fundamental connections with the mathematics of dynamical systems. Besides the heavy ball method, [Wang et al., 2021a], other examples where optimization algorithms are analyzed as dynamical systems include Nesterov momentum for smooth functions [Su et al., 2014, Muehlebach and Jordan, 2019], and more recently frameworks of analysis using Lyapunov stability theory [Lessard et al., 2016, Wilson, 2018].

By far the most widely used framework of control in optimization is that of Lyapunov's stability method, also known as Lyapunov's direct method. This method (see e.g. Slotine et al. [1991]) proposes that if the system's energy dissipates over time, then the system must be stable in some sense. The use of an energy function makes the analysis of dynamical systems more tractable, since computing the trajectories of high-dimensional or highly nonlinear systems can be cumbersome. For a given optimization algorithm, if there exists a quantity that contracts with the evolution of the algorithm, Lyapunov's direct method can be applied to show convergence. This technique has been used to analyze many existing optimization algorithms.

However, Lyapunov's direct method does not provide optimality guarantees for control: prescriptive suggestions for how to drive the system to the target state. Moreover, standard Lyapunov analysis does not take disturbances to the system into account, even though they naturally arise in most physical systems [2]. To overcome these limitations, we consider Lyapunov's second technique, the so-called "indirect method". Instead of working directly with the nonlinear system, this method studies the behavior of the system around the linearization about the equilibrium point. On one hand, this technique is more limited: the analysis only applies to linear dynamical systems. On the other hand, while it is an approach to determine stability, it is also amenable to optimal control theory. This additional power can potentially guarantee convergence to the optimal method in optimization, which is exactly our goal.

Nevertheless, optimal control has several limitations in this context of analyzing optimization algorithms. Methods for optimal control, notably LQR theory, assume knowledge of the cost functions and dynamics a priori, stochastic disturbances, and are limited to quadratic cost functions. Further, learning the optimal controller in feedback control directly is not a convex problem. We circumvent these difficulties by using new techniques in control theory devised in the context of machine learning. The online nonstochastic control framework bypasses the computational hardness issue prevalent in nonconvex problems by using convex relaxation and improper learning, leading to new regret guarantees for online control. It is applicable for an online setting, where the dynamics and the general convex costs are unknown ahead of time. Moreover, this framework provides guarantees even in the presence of adversarial disturbances – disturbances that do not follow distributional assumptions. It is very possible that optimal control theory has not been systematically applied to optimization largely due to the limitations above [3].

## B   Lyapunov's methods and their application to optimization

In this section we briefly describe how control theory, and in particular Lyapunov's direct method, has been used to analyze optimization methods. We also describe Lyapunov's indirect method, which to the best of our knowledge has not been applied to mathematical optimization extensively.

A **dynamical system** is a vector field mapping $\mathbb{R}^d$ onto itself. Using our notation, it can be written as

$$z_{t+1} = v(z_t),$$

where $v$ is the dynamics function. We use discrete time notation throughout this paper as optimization methods implemented on a computer admit discrete-time representations. Dynamical systems can be used to describe an optimization algorithm; for example, gradient descent for an objective $f$ with learning rate $\eta$ can be written as

$$x_{t+1} = x_t - \eta \nabla f(x_t),$$

---

[2] An exception is the work of Lessard et al. [2016] .

[3] see related work section.

and similarly, other iterative preconditioned gradient (or higher-order) method can be described as a dynamical system. The natural question of convergence to local or global minima, as well as the rate of convergence, can be framed as a question about the *stability* of the dynamical system. A dynamical system is said to be stable from a given starting point $x_0$ if the dynamics converges to an equilibrium from this point. There are numerous definitions of stability and equilibria of dynamical systems, and we refer the interested reader to comprehensive discussions in [Slotine et al., 1991, Hazan and Singh, 2022]. In this introductory section we consider only the most intuitive notion of convergence to a global minimum for a convex function, and the basic setting of noiseless dynamical systems in a single trajectory.

In his foundational work [Lyapunov, 1992], Lyapunov introduced two methods for certifying stability of dynamical systems.

## B.1 Lyapunov's direct method

Lyapunov's direct method involves creating an energy or potential function, called the "Lyapunov function". This function needs to be non-increasing along the trajectory of the dynamics, and strictly positive except at the equilibirum (global minimum for an optimization problem) to certify stability. A common example given in introductory courses on dynamics is that of the motion equations of the pendulum. The Lyapunov function for this system is taken to be the total energy, kinetic and potential [Tedrake, 2020].

For the discrete dynamics of gradient descent over a strongly convex objective $f$, the standard Lyapunov energy function to use is simply the Euclidean distance to optimality, or $\mathcal{E}(x) = \frac{1}{2}\|x-x^*\|^2$, where $x^*$ is the global minimizer. It can be shown that with a sufficiently small learning rate depending on the strong convexity parameter, that this energy function is monotone decreasing for the dynamics of gradient descent, showing that the system is stable [Wilson, 2018].

At this time, various other conditions on the objective function $f$, such as smoothness, convexity, and so forth, give rise to different energy functions that can prove stability, and even rates of convergence.

## B.2 Lyupanov's indirect method

Lyapunov's second approach to stability is based on the idea of linearization around the trajectory using the Taylor approximation, and then analyze the resulting linear dynamical system. More formally, let $z_0, ..., z_T$ be a given trajectory, then we can approximate the dynamics as

$$z_{t+1} = A_t z_t + w_t,$$

where $A_t$ is the Jacobian of the dynamics with respect to $z_t$ and $w_t$ is a noise that can model misspecification or other disturbance. There are several shortcomings of this approach, especially when applied to optimization, including:

1. The linearization depends on the state, i.e. $A_t$ is actually a function of $x_t$, and optimizing the trajectory over a given sequence of linear dynamical systems does not imply global optimality on the original system.

2. The linearization is a useful approximation of the dynamical system's local behavior only if the dynamics is smooth, and the time interval between measurements is small with respect to this smoothness.

These limitations might explain, at least partially, why Lyapunov's indirect method has not been used to analyze mathematical optimization algorithms. However, there are also significant advantages to this formulation. Most significantly, we can incorporate a control signal, as well as a disturbance, into the nonlinear dynamics formulation,

$$z_{t+1} = v(z_t, u_t) + w_t,$$

where $u_t$ can capture parameters of the optimization method, such as the learning rate and preconditioner. Using Lyapunov's method we can write

$$z_{t+1} = A_t z_t + B_t u_t + w_t,$$

where $A_t, B_t$ are the Jacobians with respect to the state and control. We now have a linear time-varying (LTV) dynamical system, and if the objective functions are quadratic and the disturbance is stochastic, then the optimal controller can be computed using LQR theory [Kalman, 1960].

This observation is the starting point of our investigation. Numerous challenges arise when we attempt to use this methodology to optimize the optimizer:

1. Optimal control theory requires the knowledge of all system matrices ahead of time, but in optimization, they are only determined during the optimization process.

2. Efficient algorithms for optimal control, based on the Bellman equation and backward induction, require quadratic cost functions. They are also usually less efficient than the gradient-based meta-optimization methods we consider in the paper.

3. Optimal control requires the noise to be stochastic with mean zero, and it is not robust to model misspecification, or adversarially chosen cost functions that arise in online or stochastic optimization.

We show how to overcome these challenges using the newly established framework of online nonstochastic control. The latter addresses these difficulties:

1. Online nonstochastic control does not require the knowledge of system matrices a-priori. Further, it allows adversarially chosen systems and cost functions.

2. The methods for meta-optimization we consider are themselves gradient-based and scalable. Thus we can hope to devise practical algorithms when the number of episodes increases, or the problem dimension is high.

3. Online nonstochastic control methods have strong regret guarantees under adversarially changing cost functions and systems, which naturally carries over to finite time provable regret bounds in meta-optimization.

## C Background on nonstochastic control

Our contributions build upon the recently developed methodology of online nonstochastic control (ONC). This framework applies online convex optimization to new parametrizations of classical control problems. This section gives the basic description of this framework, and a more detailed exposition appears in [Hazan and Singh, 2022].

**Problem setting.** Consider first the simple case of a linear time invariant (LTI) dynamical system in a single trajectory without resets. A linear dynamical system (LDS) evolves via the following equation:

$$z_{t+1} = Az_t + Bu_t + w_t$$

Here $z_t \in \mathbb{R}^{d_z}$ represents the state of the system, $u_t \in \mathbb{R}^{d_u}$ represents a control input and $w_t \in \mathbb{R}^{d_x}$ is a disturbance introduced to the system. The goal of the controller is to produce a sequence of control actions $u_1 \ldots u_T$ aimed at minimizing the cumulative control cost $\sum_{t=1}^{T} c_t(z_t, u_t)$. Many systems do not exhibit full observation, and a well-studied model for capturing partial observation is when the observation is a linear projection of the state, i.e.

$$y_t = Cx_t + Du_t + \xi_t,$$

where $y_t \in \mathbb{R}^{d_y}$ is the observation at time $t$ and $\xi_t \in \mathbb{R}^{d_y}$ is an additional noise term that affects the observed signal. We say that a system is fully observed if $z_t$ is observed by the controller, and usually refer to this case unless specifically stated otherwise.

The control inputs, when correctly chosen, can modify the system to induce a particular desired behavior. For example, controlling the thermostat in a data center to achieve a certain temperature, applying a force to a pendulum to keep it upright, or driving a drone to a destination.

In classical control theory the cost functions are convex quadratic and known ahead of time, and the disturbances are i.i.d. stochastic. In nonstochastic control, we instead consider a significantly broader class of general (possibly non-quadratic) convex cost functions and norm-bounded (instead of stochastic) disturbances. Both the costs and disturbances may be adversarially chosen, and only be revealed to the controller in an online fashion.

**A new objective: policy regret.** This new objective builds upon the theory of online convex optimization [Hazan et al., 2016] and regret minimization in games: instead of computing the optimal policy in a certain class, we can compete with it using improper learning via convex relaxation of the policy class. Formally, we measure the efficacy of a policy through the notion of policy regret.

$$\text{Regret} = \sum_{t=1}^{T} c_t(z_t, u_t) - \min_{\pi \in \Pi} \sum_{t=1}^{T} c_t(z_t^\pi, u_t^\pi), \tag{7}$$

where $z_t^\pi$ represents the state encountered when executing the policy $\pi$. In particular, the second term represents the total cost paid by the best (in hindsight) policy from the class $\Pi$ had we played it under the same sequence of disturbances and cost functions. In this regard, the above notion of regret is counterfactual and hence more challenging than the standard stateless notion of regret. Algorithms which achieve low policy regret are naturally adaptive, as they can perform almost as well as the best policy in the long run, regardless whether the disturbances and costs are adversarial.

But what policies are reasonable to compare against? We survey the state-of-the-art in control policies next. Then we describe new methods arising from this theory that provably compete with the strongest policy class.

## C.1 Existing and new policy classes for control

**Linear state-feedback policies.** For a matrix $K \in \mathbb{R}^{d_u \times d_z}$, we say a policy of the form $u_t = Kz_t$ is a linear state-feedback policy, or linear policy. In classical optimal control with full observation, the cost function are quadratic in the state and control, i.e.

$$c_t(z, u) = z^\top Q x + u^\top R u.$$

Under this assumption, if the system is LTI with stochastic disturbances, then the infinite-horizon optimal policy can be computed using the Bellman optimality equations (see e.g. [Tedrake, 2020]). This gives rise to the Discrete time Algebraic Riccati Equation (DARE), whose solution is the optimal policy, and it is linear. The finite-horizon optimal policy can also be derived and shown to be linear. It is thus reasonable to consider the class of all linear policies as a comparator class, especially for LTI dynamical systems. Denote the class of all linear policies as

$$\Pi_{Lin} = \{K \in \mathbb{R}^{d_z \times d_u}\}.$$

**Linear dynamical control policies.** A generalization of static state-feedback policies is that of linear dynamical controllers (LDCs). LDCs are particularly useful for partially observed LDS and maintain their own internal dynamical system according to the observations, in order to recover the hidden state of the system. A formal definition is below.

**Definition C.1** (Linear Dynamical Controller). *A linear dynamical controller $\pi$ is a linear dynamical system $(A_\pi, B_\pi, C_\pi, D_\pi)$ with internal state $s_t \in \mathbb{R}^{d_\pi}$, input $z_t \in \mathbb{R}^{d_z}$ and output $u_t \in \mathbb{R}^{d_u}$ that satisfies*

$$s_{t+1} = A_\pi s_t + B_\pi z_t, \ \ u_t = C_\pi s_t + D_\pi z_t.$$

LDCs are state-of-the-art in terms of performance and prevalence in control applications of LDS, both in the full and partial observation settings. They are known to be theoretically optimal for partially observed LDS with quadratic cost functions and normally distributed disturbances, but are more widely used in practice. Denote the class of all LDCs as

$$\Pi_{LDC} = \{A \in \mathbb{R}^{d_s \times d_s}, B \in \mathbb{R}^{d_s \times d_z}, C \in \mathbb{R}^{d_u \times d_s}, D \in \mathbb{R}^{d_u \times d_z}\}.$$

**Disturbance-feedback controllers.** An even more general class of policies is that of disturbance-feedback control, defined below. The definition we give in the main text, Definition 3.1, does not include the stabilizing controller $K$, since the system is already stable.

**Definition C.2** (Disturbance-feedback controller). *A disturbance-feedback controller (DFC) with parameters $(K, M)$ where $M = [M^1, \ldots, M^L]$ outputs control $u_t$ at state $z_t$,*

$$u_t = Kz_t + \sum_{i=1}^{H} M^i w_{t-i},$$

*where $M^i$ denotes the $i$-th matrix in $M$, instead of a matrix to its $i$-th power.*

This policy class is more general than that of LDCs and linear controllers, in the sense that for every LDS and every policy in $\Pi_{LDC}$ and $\Pi_{Lin}$, there exists a DFC that outputs exactly the same controls on the same system and sequence of disturbances. We henceforth study regret with respect to the class of DFCs, which is the most powerful of the above policy classes, and thereby giving strong performance guarantees.

**Why is regret against DFCs meaningful?** Since the class of DFCs is more general than $\Pi_{LDC}$, competing with the best DFC translates to competing with the best LDC. The latter is know to be the optimal policy for partially observed LDS with zero-mean Gaussian disturbances, a fundamental problem for control theory known as the LQG (see e.g. [Simchowitz et al., 2020]).

It follows that sublinear regret against LDCs implies near-optimality in these widely-studied theoretical settings. In addition, this guarantee is meaningful even for adversarial disturbances and general convex cost functions. Indeed, no explicit form of the optimal policy is known for general convex cost functions, and it is conjectured to be intractable by Rockafellar [1987].

### C.2   The gradient perturbation controller

The fundamental new technique introduced in [Agarwal et al., 2019] is a novel algorithm called the Gradient Perturbation Controller (GPC) for the nonstochastic control problem. For simplicity, assume that the dynamical system given by $(A, B)$ is completely known to us and the state is fully observable. Thus, given a sequence of controls and states, we can compute the corresponding sequence of disturbances.

It can be shown that directly learning the optimal linear controller $K$ is not a convex problem. However, instead of learning $K$, we can learn a sequence of matrices $\{M^i\}_{i=1}^t$ that represents the dependence of $u_t$ on $w_{t-i}$ under the execution of some linear policy. Schematically, we parameterize our policy as a DFC, $u_t = \sum_{i=1}^t M^i w_{t-i}$. Since the states of the system are linear in the past controls, and the choice of the controller ensures that the controls are linear in $\{M^i\}_{i=1}^t$, the states are also linear in $\{M^i\}_{i=1}^t$. Moreover, since the cost functions are convex in the states and controls, they are convex in $\{M^i\}_{i=1}^t$, the parameters of interest. We can thus hope to learn the parameters using standard techniques such as gradient descent, Newton's method, and so on.

However, there are two challenges with this approach. First, the number of parameters grows linearly with time, and so can the regret and running time. And second, The decision a controller makes at a particular instance affects the future through the state.

To deal with the first issue, we limit the history length of the GPC to grow very slowly with time. It can be shown that for stable (and stabilizable given a stabilizing controller) systems, a history length of $O(\log \frac{1}{\epsilon})$ is sufficient to capture the entire class of DFCs up to an $\varepsilon$ additive approximation in terms of the average cost. This logarithmic dependence of the history length on the approximation guarantee means that for the number of parameters to grow mildly, the policy regret is affected by no more than a constant factor.

The second issue is more subtle. Luckily, online learning of loss functions with memory was a topic studied before in [Anava et al., 2015]. It is shown that gradient methods guarantee near-optimal regret if the learning rate is tuned as a function of the memory length.

With all the core components in place, we provide a brief specification of the GPC algorithm in Algorithm 3 [4] . The GPC algorithm accepts as input a stabilizing controller $K$ that ensures $\rho(A + BK) < 1$, where $\rho$ is the spectral radius of the matrix $A + BK$. Such a controller $K$ can be computed for all stabilizable systems using semi-definite programming, see e.g. [Cohen et al., 2019]. The algorithm then proceeds to control using a DFC policy that is adapted to the online cost functions. Notice that the gradient of the cost function $c_t$ is taken w.r.t. the policy variables $M_t = M_t^{1:L}$. This is valid since both the control, and in turn the state, are a convex function of these variables. The notation $\prod_{\mathcal{M}}(x)$ denotes the Euclidean projection of a vector $x$ onto the set $\mathcal{M}$, see [Hazan et al., 2016] for more details on projections.

The GPC algorithm comes with a near-optimal regret guarantee vs. the class of DFC policies,

---

[4]where $c_t(M)$ is the truncated cost as a function of a fixed DFC policy $M$, for details see [Agarwal et al., 2019].

---

**Algorithm 3** Gradient perturbation controller

1: **Input:** Step size schedule $\eta_t$, history length $L$, constraint set $\mathcal{M}$, stabilizing controller $K$.
2: Initialize $M^{1:L} \in \mathcal{M}$ arbitrarily.
3: **for** $t = 1, \dots, T$ **do**
4:     Choose the action: $u_t = K z_t + \sum_{l=1}^{L} M_t^l w_{t-l}$.
5:     Observe the new state $z_{t+1}$, cost function $c_t$, and record $w_t = z_{t+1} - A z_t - B u_t$.
6:     Online gradient update:
$$M^{t+1} = \Pi_{\mathcal{M}}(M^t - \eta_t \nabla c_t(M^t)).$$

7: **end for**

---

**Theorem C.3** (Theorem 5.1 in [Agarwal et al., 2019]). *Let $u_t$ be a sequence of controls generated by Algorithm 3 for a known LDS. Then for any arbitrary bounded disturbance sequence and convex cost functions, it holds that*

$$\sum_{t=1}^{T} c_t(z_t, u_t) - \min_{\pi \in \Pi_{DFC}} \sum_{t=1}^{T} c_t(z_t^\pi, u_t^\pi) = \tilde{O}(\sqrt{T}).$$

### C.3 Significance of online nonstochastic control to meta-optimization

**Significance of online nonstochastic control theory for online linear control.** Online nonstochastic control theory yields the first provably efficient method for LQR under general cost functions and adversarial disturbances with finite time guarantees, which was an important open problem proposed by [Rockafellar [1987]]. The LQR problem is a fundamental one, and a building block for more sophisticated techniques in model predictive control.

**Significance with respect to meta optimization.** We formulate meta-optimization as a control problem with adversarial disturbances. Furthermore, our goal is to minimize meta-regret which is an adversarial notion with respect to the best policy in hindsight.

Regret minimization with nonstochastic disturbances was not known before the introduction of the online nonstochastic control framework, and the new methods therein. In formulating meta-optimization as a control problem, we make use of the crucial fact that these methods can tolerate adversarial disturbances, which are chosen online along a path of optimization.

## D Smooth convex meta-optimization

In this section, we present the details for meta-optimization when the objective functions are not quadratic, but smooth and convex functions satisfying the following assumptions.

**Assumption 4.** *The objective functions $f_{t,i}$ have uniformly bounded Hessians,*

$$\|\nabla^2 f_{t,i}(x)\| \le \beta, \ \ \forall \, x, \, t, \, i.$$

In particular, the assumption above implies that he objective functions have Lipschitz gradients,

$$\|\nabla f_{t,i}(x) - \nabla f_{t,i}(y)\| \le \beta \|x - y\|,$$

and

$$\|\nabla f_{t,i}(x)\| \le \beta \|x\| + b \tag{8}$$

for some $b \ge 0$, for all $x, t, i$.

### D.1 The dynamics of smooth convex meta-optimization

For smooth convex meta-optimization, we can define the system evolution along the trajectory of the iterates that are actually played during meta-optimization. Consider the gradient of $f_{t,i}$, $\nabla f_{t,i} \in \mathbb{R}^d$, and let $\nabla f_{t,i}^j(x)$ denote the $i$-th coordinate of the gradient. Let $\nabla^2 f_{t,i}^j(x) = \frac{\partial \nabla f_{t,i}^j(x)}{\partial x} \in \mathbb{R}^d$ be the

gradient of $\nabla f_{t,i}^j(x)$, and define

$$H_{t,i}(y_1, \ldots, y_d) = \begin{bmatrix} \nabla^2 f_{t,i}^1(y_1) \\ \vdots \\ \nabla^2 f_{t,i}^d(y_d) \end{bmatrix}.$$

The system evolution is

$$\begin{bmatrix} x_{t+1,i} \\ x_{t,i} \\ \nabla f_{t,i}(x_{t,i}) \end{bmatrix} = \begin{bmatrix} (1-\delta)I & 0 & -\eta I \\ I & 0 & 0 \\ H_{t,i} & -H_{t,i} & 0 \end{bmatrix} \times \begin{bmatrix} x_{t,i} \\ x_{t-1,i} \\ \nabla f_{t-1,i}(x_{t-1,i}) \end{bmatrix} + \begin{bmatrix} I & 0 & 0 \\ 0 & 0 & 0 \\ 0 & 0 & 0 \end{bmatrix} \times u_{t,i} + \begin{bmatrix} 0 \\ 0 \\ \nabla f_{t,i}(x_{t-1,i}) \end{bmatrix},$$

$$(9)$$

where $H_{t,i}$ satisfies

$$H_{t,i} = H_{t,i}(\xi_{t,i}^1, \ldots, \xi_{t,i}^d)$$

for some $\xi_{t,i}^j$ on the line segment from $x_{t-1,i}$ to $x_{t,i}$, for all $j \in [d]$. By the mean value theorem, we can find $\{\xi_{i,t}^j\}_{j=1}^d$ such that

$$\nabla f_{t,i}(x_{t,i}) = \nabla f_{t,i}(x_{t-1,i}) + H_{t,i}(x_{t,i} - x_{t-1,i}).$$

Further, we make the following assumption on $H_{t,i}$ uniformly. In the case of smooth quadratic objective functions, $H_{t,i}$ is the Hessian, and the assumption is satisfied.

**Assumption 5.** *For all $(t, i)$, $\rho(H_{t,i}) \le \beta$.*

Note that $H_{t,i}$ is in fact not directly observable to us. **Crucially, however, we do not need to use $H_{t,i}$ or any system information in our algorithm**; all we need to know are the disturbances, which can be computed by taking gradients of the objective functions.

**Resets.** The reset disturbance is the same, repeated here for completeness,

$$w_{T,i} = \begin{bmatrix} x_{1,i+1} - ((1-\delta)x_{T,i} - \eta \nabla f_{T-1,i}(x_{T-1,i}) + \bar{u}_{T,i}) \\ x_{1,i+1} - x_{T,i} \\ \nabla f_{T,i}(x_{T-1,i}) - \nabla f_{T,i}(x_{T,i}) \end{bmatrix},$$

$$(10)$$

where $\bar{u}_{T,i}$ is the top $d$ entries of the control signal $u_{T,i}$. We again assume that the starting point in each epoch has bounded norm by Assumption 2.

**Cost functions.** We again consider minimizing the function value: $c_{t,i}(z_{t,i}, u_{t,i}) = f_{t,i}(Sz_{t,i}) = f_{t,i}(x_{t,i})$, where $S$ is a matrix that selects the first $d$ entries of $z_{t,i}$.

**Stability.** We consider problems where Assumption 3 holds on the linearized dynamics.

## D.2 Algorithm

Let $\mathcal{M}_{\delta_M}$ denote the Minkowski subset of $\mathcal{M}$: $\mathcal{M}_{\delta_M} = \{M \in \mathcal{M} : \frac{1}{1-\delta_M}M \in \mathcal{M}\}$, consider the following bandit algorithm for smooth convex meta-optimization:

---

**Algorithm 4** Smooth convex meta-optimization

---

1: **Input:** $N, T, z_{1,1}, \eta, \delta, \{\eta^g_{t,i}\}, L, \delta_M$, starting points $\{x_{1,i}\}_{i=1}^N, \kappa, \gamma$
2: **Set:** $\mathcal{M} = \{M = \{M^1, \ldots, M^L\} : \|M^l\| \le \kappa^3(1-\gamma)^l\}$.
3: Initialize any $M_{1,1} = \cdots = M_{L,1} \in \mathcal{M}_{\delta_M}$.
4: Sample $\epsilon_{1,1}, \ldots, \epsilon_{L,1} \in_{\mathbb{R}} \mathbb{S}_1^{L \times 3n \times 3n}$, set $\widetilde{M}_{l,1} = M_{l,1} + \delta_M \epsilon_{l,1}$ for $l = 1, \ldots, L$.
5: **for** $i = 1, \ldots, N$ **do**
6:     If $i > 1$, set $z_{1,i} = z_{T+1,i-1}, M_{1,i} = M_{T+1,i-1}$.
7:     **for** $t = 1, \ldots, T$ **do**
8:         Choose $u_{t,i} = \sum_{l=1}^L \widetilde{M}^l_{t,i} w_{t-l,i}$.
9:         Receive $f_{t,i}$, compute $\nabla f_{t,i}(x_{t,i}), \nabla f_{t,i}(x_{t-1,i})$. If $t = T$, then compute $w_{T,i}$ according to 4. // Now we have $z_{t+1,i}, w_{t,i}$.
10:         Suffer control cost $c_{t,i}(z_{t,i}) = f_{t,i}(x_{t,i})$.
11:         Store $g_{t,i} = \dfrac{9n^2 L}{\delta_M} c_{t,i}(z_{t,i}) \sum_{l=1}^L \epsilon_{t-l,i}$ if $t \ge L$ else 0.
12:         Perform gradient update on the controller parameters: $M_{t+1,i} = \Pi_{\mathcal{M}_{\delta_M}} (M_{t,i} - \eta^g_{t,i} g_{t-L,i})$.
13:         Sample $\epsilon_{t+1,i} \in_{\mathbb{R}} \mathbb{S}_1^{L \times 3n \times 3n}$, set $\widetilde{M}_{t+1,i} = M_{t+1,i} + \delta_M \epsilon_{t+1,i}$
14:     **end for**
15: **end for**

---

The algorithm uses noise-based gradient estimators for the gradients with respect to $M^l_{t,i}$. At each iteration, a noise is sampled in the unit sphere (Line 13) and added to the updated iterate of $M_{t+1,i}$. Then, gradient estimators are constructed on Line 11 by using cost function evaluations, and an update step is performed on Line 12.

## E   General dynamical system formulation

The most general formulation of the dynamical system for meta-optimization can have a number of past states and gradients. For any given $h \le T/2$, consider the time-variant, discrete linear dynamical system of dimension $2hd$ as follows

$$
z_{t+1,i} = \begin{bmatrix} x_{t+1,i} \\ \vdots \\ x_{t-h+2,i} \\ \nabla f_{t,i}(x_{t,i}) \\ \vdots \\ \nabla f_{t-h+1,i}(x_{t-h+1,i}) \end{bmatrix} = A_{t,i} \times \begin{bmatrix} x_{t,i} \\ \vdots \\ x_{t-h+1,i} \\ \nabla f_{t-1,i}(x_{t-1,i}) \\ \vdots \\ \nabla f_{t-h,i}(x_{t-h,i}) \end{bmatrix} + B \times u_{t,i} + \begin{bmatrix} 0 \\ \vdots \\ 0 \\ \nabla f_{t,i}(x_{t-1,i}) \\ 0 \\ \vdots \\ 0 \end{bmatrix},
$$

where

$$
A_{t,i} = \begin{bmatrix} (1-\delta)I & 0 & \cdots & \cdots & 0 & -\eta I_d & 0 & \cdots & 0 \\ I_{d(h-1)} & & & & & 0 & 0 & \cdots & \cdots & 0 \\ H_{t,i} & -H_{t,i} & 0 & \cdots & 0 & 0 & 0 & \cdots & 0 \\ 0 & \cdots & \cdots & \cdots & 0 & I_{d(h-1)} & & & 0 \end{bmatrix}, \quad B = \begin{bmatrix} I_d & 0 & \cdots & 0 \\ 0 & 0 & \cdots & 0 \\ & & \vdots & \\ 0 & 0 & \cdots & 0 \end{bmatrix}.
$$

(11)

This formulation holds for a similar reason to the simplified dynamical system (3), namely the relation

$$
\nabla f_{t,i}(x_{t,i}) = \nabla f_{t,i}(x_{t-1,i}) + H_{t,i}(x_{t,i} - x_{t-1,i}).
$$

In the deterministic setting, more expressive states can lead to a richer benchmark algorithm class and a stronger guarantee. We have an LTI system in the deterministic setting, and the class of DFCs can compete with the class of linear controllers, see Appendix C. In this more general formulation, the states contain gradients and iterates further into the past, so the class of linear controllers correspond to optimization algorithms that can use more past gradients and iterates. This benchmark algorithm class can be potentially larger than the formulation considered in Section 3.

# F    Optimization settings

Our main theorem can be refined for each of the meta-optimization settings described in Section 1.1.

## F.1    Deterministic optimization

Consider the setting where we receive the same quadratic function in each epoch, i.e. $f_{t,i} = f$ for all $t, i$. In this case, we have an LTI system, and we can obtain stronger guarantees compared to LTV systems. First, the sequential stability assumption on the system can be simplified to a standard strong stability assumption.

**Assumption 6.** *The system dynamics are $(\kappa, \gamma)$-strongly stable, with $\kappa \geq 1$.*

A fundamental class of policies considered in control theory is the set of linear state-feedback policies. It is well-understood that for certain classical control problems, such as the linear quadratic regulator, linear policies are optimal. We give the formal definition of stable linear policies below.

**Definition F.1** (Strongly stable linear policies)**.** *Given a system with dynamics $(A, B)$, a policy $K$ is linear if $u_t = K z_t$. A linear policy is $(\kappa, \gamma)$-strongly stable if the dynamics $A + BK$ is $(\kappa, \gamma)$ strongly stable, and $\|K\| \leq \kappa$.*

**Remark F.2.** *It is natural to consider the relationship between policy classes. We say a class of policies $\Pi_1$ approximates another class $\Pi_2$ if for all systems, for any $\pi_2 \in \Pi_2$, there exists $\pi_1 \in \Pi_1$ that incur average cost close to $\pi_2$. As shown in Agarwal et al. [2019], for stable LTI systems, $\Pi_{DFC}$ can approximate the class of strongly stable linear policies, denoted as $\Pi_{lin}^{\kappa, \gamma}$. Therefore, we can consider $\Pi_{lin}^{\kappa, \gamma}$ as the benchmark policy class in this setting. The corresponding benchmark algorithm class $\Pi$ is given below.*

In the deterministic setting, we consider optimization algorithms parameterized by a matrix $K = [K_1 \; K_2 \; K_3] \in \mathbb{R}^{d \times 3d}$. Let $x_{t,i}$ be the iterates played by our algorithm, and

$$z_{t,i}^K = \begin{bmatrix} x_{t,i}^K \\ x_{t-1,i}^K \\ \hat{\nabla} f(x_{t-1,i}^K) \end{bmatrix}, \quad \text{where } \hat{\nabla} f(x_{t-1,i}^K) = \nabla f(x_{t-1,i}^K) - \nabla f(x_{t-2,i}^K) + \nabla f(x_{t-2,i}),$$

be the state at time $(t, i)$ reached by the optimizer parameterized by $K$. The optimizer with parameter $K$ has the corresponding updates:

$$x_{t+1,i}^K = ((1 - \delta)I + K_1)x_{t,i}^K + K_2 x_{t-1,i}^K + (K_3 - \eta I)\hat{\nabla} f(x_{t-1,i}^K). \tag{12}$$

This class $\Pi$ can capture common optimization algorithms on time-delayed pseudo-gradients $\hat{\nabla} f$ for the deterministic setting, and we give some examples in the following table. For more details on the class of algorithms and range of hyperparameters, see Appendix G; for a more concrete example of learning the learning rate, see Appendix H.

| Method | K | Update |
|---|---|---|
| GD with learning rate $\eta'$ | $[0 \quad 0 \quad (\eta - \eta')I]$ | $x_{t+1,i}^K = (1 - \delta)x_{t,i}^K - \eta'\hat{\nabla} f(x_{t-1,i})$ |
| Momentum | $[-\beta I \quad \beta I \quad 0]$ | $x_{t+1,i}^K = (1 - \delta - \beta)x_{t,i}^K + \beta x_{t-1,i}^K - \eta\hat{\nabla} f(x_{t-1,i})$ |
| Preconditioned methods | $[0 \quad 0 \quad \eta I - P]$ | $x_{t+1,i}^K = (1 - \delta)x_{t,i}^K - P\hat{\nabla} f(x_{t-1,i})$ |

The benchmark algorithm class we consider also includes any combination of the above methods, which can be expressed by an appropriate choice of $K$.

Let $\bar{x} = \frac{1}{TN} \sum_{i=1}^N \sum_{t=1}^T x_{t,i}$ be the average iterate, and $\bar{J}(\mathcal{A}) = \frac{1}{TN} \sum_{i=1}^N J_i(\mathcal{A})$ denote the average cost of the algorithm $\mathcal{A}$. Then by convexity, Theorem 3.2 implies

$$f(\bar{x}) \; \leq \; \min_{\mathcal{A} \in \Pi} \bar{J}(\mathcal{A}) + \tilde{O}\left(\frac{1}{\sqrt{TN}}\right).$$

## F.2 Stochastic optimization

In this setting, our functions are drawn randomly from distributions $\mathcal{D}_1, \mathcal{D}_2, \ldots, \mathcal{D}_N$ that vary from epoch to epoch. In epoch $i$, for each time step $t \in [T]$, we draw a quadratic function $f_{t,i} \sim \mathcal{D}_i$. Let $\mathbb{E}$ denote the unconditional expectation with respect to the randomness of the functions, and define the function $\bar{f}_i(x) := \mathbb{E}_{\mathcal{D}_i}[f_{t,i}(x)]$, then the guarantee can be written as

$$\frac{1}{NT} \sum_{i=1}^{N} \sum_{t=1}^{T} \mathbb{E}[\bar{f}_i(x_{t,i})] \leq \frac{1}{NT} \min_{\mathcal{A} \in \Pi} \sum_{i=1}^{N} \sum_{t=1}^{T} \mathbb{E}\left[\bar{f}_i(x_{t,i}^{\mathcal{A}})\right] + \tilde{O}\left(\frac{1}{\sqrt{TN}}\right).$$

## F.3 Adversarial online optimization

Consider the setting where we have a new function at each time step of each epoch in the optimization process. These functions can arrive in an online and adversarial manner; in other words, they do not satisfy distributional assumptions such as ones presented in the last subsection. This setting describes the meta-online convex optimization (meta-OCO) problem, and we give our guarantees in the standard OCO metric – regret.

Let $x_i^* = \arg\min_x \sum_{t=1}^{T} f_{t,i}(x)$ be the optimum in hindsight in episode $i$, and denote $\text{Regret}_i(\mathcal{A})$ as the regret suffered by the algorithm $\mathcal{A}$ in epoch $i$. Subtracting $\sum_{i=1}^{N} \sum_{t=1}^{T} f_{t,i}(x_i^*)$ on both sides,

$$\frac{1}{TN} \sum_{i=1}^{N} \text{Regret}_i \leq \min_{\mathcal{A} \in \Pi} \frac{1}{TN} \sum_{i=1}^{N} \text{Regret}_i(\mathcal{A}) + \tilde{O}\left(\frac{1}{\sqrt{NT}}\right),$$

that is, over episodes, the average regret approaches that of the best online learner in the class $\Pi$.

# G   Expressing optimization algorithms as policies

In this section, we give more details on algorithms that can be captured by the policy class we consider in the deterministic setting, namely linear stabilizing policies. Let $K$ be a stabilizing policy, then it must satisfy

$$\rho(A + BK) < 1,$$

where $A$ is the system dynamics. We can write

$$A + BK = \begin{bmatrix} (1-\delta)I + K_1 & K_2 & -\eta I + K_3 \\ I & 0 & 0 \\ H & -H & 0 \end{bmatrix}.$$

If $\lambda$ is an eigenvalue of $A + BK$, then

$$\det\left(\begin{bmatrix} (1-\delta-\lambda)I + K_1 & K_2 & -\eta I + K_3 \\ I & -\lambda I & 0 \\ H & -H & -\lambda I \end{bmatrix}\right) = 0.$$

We can compute the determinant by methods developed in Powell [2011]. Let

$$M_1 = (1-\delta-\lambda)I + K_1 + \frac{(K_3 - \eta I)H}{\lambda},$$

$$M_2 = K_2 - \frac{(K_3 - \eta I)H}{\lambda},$$

then $\det(A+BK-\lambda I) = \det(M_1 + \frac{M_2}{\lambda})\det(-\lambda I)^2$. For $\lambda \neq 0$, this implies that $\det(M_1 + \frac{M_2}{\lambda}) = 0$. Expanding the expression, we have

$$\det\left((1-\delta-\lambda)I + K_1 + \frac{\lambda-1}{\lambda^2}(K_3 - \eta I)H + \frac{K_2}{\lambda}\right) = 0,$$

suggesting that $\lambda + \delta - 1$ is an eigenvalue of $K_1 + \frac{\lambda-1}{\lambda^2}(K_3 - \eta I)H + \frac{K_2}{\lambda}$. We show in the following subsections that non-trivial algorithms can be expressed as stabilizing linear policies, by upper bounding $|\lambda|$ using this relationship.

## G.1 GD with fixed learning rates

We can take $K_1 = K_2 = 0$, $K_3 = \eta' I$ to encode GD with learning rate $\eta' - \eta$. By the following lemma, any $|\eta' - \eta| \leq 1/8\beta$ is a stabilizing linear policy, where $\beta = \|H\|$.

**Lemma G.1.** *Suppose the conditions in Lemma I.2 are satisfied. Let $K_3 = \eta' I$, then for $\eta'$ such that $|\eta' - \eta| \leq 1/8\beta$, for any $\lambda \in \mathbb{C}$ where $\lambda + \delta - 1$ is an eigenvalue of $\frac{\lambda - 1}{\lambda^2}(\eta' - \eta)H$, we have $|\lambda| < 1 - \delta/2$.*

*Proof.* The proof is similar to the proof of Lemma I.2. Let $\Sigma_{ii}$ denote the $i$-th eigenvalue of $H$, then $\lambda$ must satisfy, for some $i$,

$$\frac{\lambda - 1}{\lambda^2}(\eta' - \eta)\Sigma_{ii} = \lambda + \delta - 1.$$

Rearranging and taking the absolute value, we obtain

$$|\eta - \eta'|\Sigma_{ii}\delta = |\lambda^2 - (\eta' - \eta)\Sigma_{ii}||\lambda - 1 + \delta|.$$

Suppose $|\lambda| \geq 1 - \delta/2$, then

$$|\eta - \eta'|\Sigma_{ii}\delta \geq (|\lambda|^2 - |\eta' - \eta|\Sigma_{ii})(|\lambda| - (1 - \delta))$$
$$\geq (|\lambda|^2 - |\eta' - \eta|\Sigma_{ii})\delta/2,$$

and $2|\eta - \eta'|\Sigma_{ii} \geq |\lambda|^2 - |\eta' - \eta|\Sigma_{ii} \Rightarrow |\eta - \eta'|\Sigma_{ii} \geq (1 - \delta/2)^2/3$. Since $|\eta - \eta'| \leq 1/8\beta$, $|\eta - \eta'|\Sigma_{ii} \leq 1/8$, while the right hand side is at least $3/16$, and we have a contradiction. $\square$

## G.2 Momentum

In this case, $K_3 = 0$, $K_1 = -vI$, $K_2 = vI$ describes momentum with parameter $v$, and we show for $v \leq \delta$, the corresponding linear policy is stabilizing.

**Lemma G.2.** *Suppose $\eta, \delta$ satisfy the conditions in Lemma I.2. Then for $v \in [0, \delta]$, for any $\lambda \in \mathbb{C}$ where $\lambda + \delta - 1$ is an eigenvalue of $-vI - \frac{\eta(\lambda - 1)}{\lambda^2}H + \frac{v}{\lambda}I$, we have $|\lambda| < 1 - \delta/4$.*

*Proof.* Let $\Sigma_{ii}$ denote the $i$-th eigenvalue of $H$, and let $c_i = \eta\Sigma_{ii}$. Then $\lambda$ must satisfy, for some $i$,

$$\frac{v}{\lambda} - v - \frac{c_i(\lambda - 1)}{\lambda^2} = \lambda + \delta - 1.$$

Rearranging and taking the absolute value, we obtain

$$|\lambda^2 - v + c_i||\lambda - (1 - \delta - v)| = |v - (\delta + v)(v - c_i)|.$$

Since $0 \leq v \leq 1/2$, the right hand side is equal to $v - (\delta + v)(v - c_i)$. This can be seen as follows: if $v \leq c_i$, then the statement is true; if $v \geq c_i$, then $v \geq v - c_i \geq (\delta + v)(v - c_i)$. Assume $|\lambda| \geq 1 - \delta/4$, and we show the lemma by contradiction. We first write,

$$c_i + (1 - \delta - v)(v - c_i) \geq |\lambda^2 - v + c_i||\lambda - (1 - \delta - v)| \geq (|\lambda|^2 - |v - c_i|)(|\lambda| - (1 - \delta - v))$$
$$\geq ((1 - \delta/4)^2 - |v - c_i|)((1 - \delta/4) - (1 - \delta - v))$$

If $v \leq c_i$, the expression becomes

$$c_i + 2(1 - \delta - v)(v - c_i) + (c_i - v)(1 - \frac{\delta}{4}) = 2(\delta + v)(c_i - v) + v - \frac{\delta}{4}(c_i - v) \geq (1 - \frac{\delta}{4})^2(\frac{3\delta}{4} + v).$$

Note that the left hand side is upper bounded by $\frac{\delta + v}{4} + v$ because $c_i \leq 1/8$, and we have

$$\frac{\delta}{4} + \frac{5v}{4} \geq \frac{49}{64}(\frac{3\delta}{4} + v),$$

which is a contradiction, because $v \leq \delta$. Now, suppose $v > c_i$, we have

$$c_i + (v - c_i)(1 - \delta/4) + (1 - \delta - v)(1 - \delta/4)^2 \geq (1 - \delta/4)^3.$$

We upper bound the left hand side using $1 \geq 1 - \delta/4$, and obtain $1 - \delta \geq (1 - \delta/4)^3$, which is a contradiction for $\delta \in [0, 1/2)$. $\square$

### G.3 Preconditioned methods

Similar to the learning rate case, we set $K_1 = K_2 = 0$, and $K_3 = \eta I - P$, where $P$ is the preconditioner. The following lemma shows that for $P$ such that $\rho(PH) \leq 1/8$, the linear policy specified by $K_1, K_2, K_3$ is stabilizing.

**Lemma G.3.** *Suppose $\eta, \delta$ satisfy the conditions in Lemma I.2. Then for $P$ such that $\rho(PH) \leq 1/8$, for any $\lambda \in \mathbb{C}$ where $\lambda + \delta - 1$ is an eigenvalue of $-\frac{\lambda-1}{\lambda^2} PH$, we have $|\lambda| < 1 - \delta/2$.*

*Proof.* Let $c_i$ denote the $i$-th eigenvalue of $PH$. Then for some $i$,

$$\frac{1-\lambda}{\lambda^2} c_i = \lambda + \delta - 1.$$

Assume $|\lambda| \geq 1 - \delta/2$. After algebraic manipulation and taking the absolute value, we have

$$|\lambda^2 + c_i||\lambda + \delta - 1| = \delta|c_i| \geq (|\lambda|^2 - |c_i|)(|\lambda| - (1-\delta)) \geq (|\lambda|^2 - |c_i|)\delta/2.$$

The above inequality implies that $3|c_i| \geq |\lambda|^2 \geq 9/16$, which is a contradiction, since $|c_i| \leq 1/8$ by definition. $\square$

## H Example: learning the learning rate for convex quadratics

Consider the deterministic setting , where we receive a quadratic objective function $f(x) = \frac{1}{2} x^\top H x$ that remains invariant over the course of meta-optimization. Assume $\|H\| = \beta \geq 1$. For gradient descent, a good choice of learning rate is $\frac{1}{\beta}$, but often we only have an upper bound $\hat{\beta}$ such that $\hat{\beta} \gg \beta$. As we show in the sequel, we can do almost as well as gradient descent with learning rate $\frac{1}{8\beta}$ on average using meta-optimization, which is a constant factor away.

Suppose we choose $\eta_g, L, \delta$ according to Theorem 3.2, and set $\eta = \frac{1}{8\hat{\beta}}$. Then by Lemma I.2, the dynamical system is stable, and Assumptions 6,2,1 are satisfied. By Theorem 3.2, we can compete with the best stabilizing linear policy. For a linear policy $K$, let

$$z_{t,i}^K = \begin{bmatrix} x_{t,i}^K \\ x_{t-1,i}^K \\ \hat{\nabla} f^K(x_{t-1,i}^K) \end{bmatrix}$$

denote the state reached at time $(t, i)$ by playing policy $K$. Let $[K_1 \ K_2 \ K_3] \in \mathbb{R}^{d \times 3d}$ represent the top $d$ rows of $K$, where the submatrices have dimension $d \times d$. The closed-loop dynamics of the linear policy $K$ is

$$z_{t+1,i}^K = \begin{bmatrix} (1-\delta)I + K_1 & K_2 & -\frac{1}{8\hat{\beta}}I + K_3 \\ I & 0 & 0 \\ H_{t,i} & -H_{t,i} & 0 \end{bmatrix} z_{t,i}^K + w_{t,i}.$$

Setting $K_1 = 0$, $K_2 = 0$, $K_3 = -(\frac{1}{8\beta} - \frac{1}{8\hat{\beta}})I$, the dynamics is time-delayed gradient descent (using pseudo-gradients) with learning rate $\frac{1}{8\beta}$ and weight decay. By Lemma I.2, the closed-loop dynamics is stable, so $K$ is a stabilizing linear policy and we do at least as well as playing $K$ on average.

## I System stability

In this section, we discuss the stability of the system in our dynamical systems formulation under the deterministic setting. Since the system is LTI, only strong stability is required for applying nonstochastic control, instead of sequential stability. We start with the usual definition of a stable system.

A system $x_{t+1} = Ax_t + Bu_t + w_t$ is said to be stable if the spectral radius of $A$, denoted as $\rho(A)$, is bounded away from 1. The following lemma shows that if a system is stable, it is also strongly stable with some parameters $(\kappa, \gamma)$.

**Lemma I.1** (Lemma B.3 in Cohen et al. [2018]). *If the system $A$ is stable with $\rho(A) < 1 - \gamma$ for some $\gamma > 0$, it is also $(\kappa, \gamma)$-strongly stable, where $\kappa = \max\{\|P\|, \|P^{-1}\|\}$ with $P = \sum_{i=0}^{\infty} (A^i)^{\top} A^i$.*

Therefore, we focus on showing that the dynamical system we consider is stable in the usual sense. The lemma below shows that if we set the learning rate of the base gradient descent dynamics to be sufficiently small, then the system is stable without control signals. This restriction on $\eta$ is natural since we expect gradient descent to converge with step size smaller than $1/\beta$.

**Lemma I.2.** *Consider the dynamical system formulation (3), and suppose $0 \preceq H_{t,i} \preceq \beta I$, then for $\eta \leq \frac{1}{8\beta}, \delta \in (0, \frac{1}{2}]$, we have*

$$\rho\left(\begin{bmatrix} (1-\delta)I & 0 & -\eta I \\ I & 0 & 0 \\ H_{t,i} & -H_{t,i} & 0 \end{bmatrix}\right) < 1 - \frac{\delta}{2} < 1.$$

*Proof of Lemma I.2.* Let $A = \begin{bmatrix} (1-\delta)I & 0 & -\eta I \\ I & 0 & 0 \\ H & -H & 0 \end{bmatrix}$ By definition, if $\lambda$ is an eigenvalue of $A$,

then

$$\det\left(\begin{bmatrix} (1-\delta-\lambda)I & 0 & -\eta I \\ I & -\lambda I & 0 \\ H & -H & -\lambda I \end{bmatrix}\right) = 0.$$

We can then use Section 4.2 of Powell [2011] to compute the determinant of $A - \lambda I$. Write $A - \lambda I = \begin{bmatrix} S_{11} & S_{12} & S_{13} \\ S_{21} & S_{22} & S_{23} \\ S_{31} & S_{32} & S_{33} \end{bmatrix}$, we have

$$S_{11} - S_{13}S_{33}^{-1}S_{31} = (1-\delta-\lambda)I - (-\eta I)(-\frac{1}{\lambda}I)H = (1-\delta-\lambda)I - \frac{\eta}{\lambda}H.$$

$$S_{12} - S_{13}S_{33}^{-1}S_{32} = 0 - (-\eta I)(-\frac{1}{\lambda}I)(-H) = \frac{\eta}{\lambda}H.$$

$$S_{22} - S_{23}S_{33}^{-1}S_{32} = -\lambda I.$$

$$S_{21} - S_{23}S_{33}^{-1}S_{31} = I.$$

By Equation 4.8 in Powell [2011],

$$\det(A - \lambda I) = \det((1-\delta-\lambda)I - \frac{\eta}{\lambda}H - \frac{\eta}{\lambda}H(-\frac{1}{\lambda}I))\det(-\lambda I)^2$$
$$= \det((1-\delta-\lambda)I - \frac{\eta}{\lambda}H + \frac{\eta}{\lambda^2}H)\det(-\lambda I)^2.$$

Therefore, if $\lambda$ is an eigenvalue of $A$, it must hold that $\det((1-\delta-\lambda)I - \frac{\eta}{\lambda}H + \frac{\eta}{\lambda^2}H) = 0$. Let $H = U\Sigma U^{\top}$ be the eigenvalue decomposition of $H$. Since

$$\det((1-\delta-\lambda)I - \frac{\eta}{\lambda}H + \frac{\eta}{\lambda^2}H) = \prod_{i=1}^{d}(1-\lambda-\delta - \frac{\eta}{\lambda}\Sigma_{ii} + \frac{\eta}{\lambda^2}\Sigma_{ii}),$$

it follows that for some $i$,

$$1 - \lambda - \delta - \frac{\eta}{\lambda}\Sigma_{ii} + \frac{\eta}{\lambda^2}\Sigma_{ii} = 0.$$

Let $\eta\Sigma_{ii} = c_i$, and by our choice of $\eta$, $|c_i| \leq \frac{1}{8}$ for all $i \in [d]$. We can re-write the above cubic equation as

$$\lambda^3 - (1-\delta)\lambda^2 + c_i\lambda - c_i = 0,$$

and we will prove the lemma by contradiction. First, observe that $\lambda^3 - (1-\delta)\lambda^2 + c_i\lambda - c_i = 0 \Rightarrow (\lambda^2 + c_i)(\lambda - 1 + \delta) = \delta c_i$. Suppose $|\lambda| \geq 1 - \delta/2 \geq 3/4$. By triangle inequality of the complex modulus, $|\lambda - 1 + \delta| \geq |\lambda| - |1 - \delta| \geq \delta/2$. Since $|\lambda^2 + c_i||\lambda - 1 + \delta| = \delta|c_i| \geq |\lambda^2 + c_i|\delta/2$, it must be that $|c_i| \geq |\lambda^2 + c_i|/2 \geq (|\lambda|^2 - |c_i|)/2$, and $3|c_i| \geq |\lambda|^2 \geq 9/16$, which is a contradiction. $\square$

We also give an analogous lemma for the general system formulation given in (11).

**Lemma I.3.** *Suppose* $0 \preceq H_t \preceq \beta I$, *then for* $\delta \in (0, \frac{1}{2}]$, $\eta < \frac{\delta}{16\beta}$, *for* $A_{t,i}$ *defined in* 11,

$$\rho(A_{t,i}) < 1 - \frac{\delta}{2} < 1.$$

*Proof.* We again use Powell [2011] to determine the eigenvalues of the dynamics matrix. First, we decompose $A_t$ into blocks and write

$$A_t - \lambda I = \begin{bmatrix} S_{11} & S_{12} \\ S_{21} & S_{22} \end{bmatrix},$$

where $S_{ij} \in \mathbb{R}^{dh \times dh}$. Then we have

$$\det(A_t - \lambda I) = \det(S_{11} - S_{12}S_{22}^{-1}S_{21})\det(S_{22}).$$

Since $S_{22}$ is a lower triangular matrix with $-\lambda$ on the diagonal, $\det(S_{22}) = 0$ if and only if $\lambda = 0$. Thus for $\lambda \neq 0$, $\det(A_t - \lambda I) = 0$ if and only if $\det(S_{11} - S_{12}S_{22}^{-1}S_{21}) = 0$. Now, let $S_{22}^{-1} = \begin{bmatrix} A & 0 \\ C & D \end{bmatrix}$ where $A \in \mathbb{R}^{d \times d}$, and $D \in \mathbb{R}^{d(h-1) \times d(h-1)}$. Then

$$S_{12}S_{22}^{-1}S_{21} = \begin{bmatrix} -\eta A & 0 \\ 0 & 0 \end{bmatrix} S_{21} = \begin{bmatrix} -\eta A H_t & \eta A H_t & 0 \cdots 0 \\ 0 & \cdots & 0 \cdots 0 \end{bmatrix}$$

$$= \begin{bmatrix} \frac{\eta}{\lambda} H_t & -\frac{\eta}{\lambda} H_t & 0 \cdots 0 \\ 0 & \cdots & 0 \cdots 0 \end{bmatrix} \in \mathbb{R}^{dh \times dh}$$

where the last equality holds because $A = -\frac{1}{\lambda} I_d$.

$$S_{11} - S_{12}S_{22}^{-1}S_{21} = \begin{bmatrix} (1 - \delta - \lambda)I_d & 0 & 0 & \cdots & 0 \\ I_d & -\lambda I_d & 0 & \cdots & 0 \\ 0 & I_d & -\lambda I_d & 0 \cdots & 0 \\ \vdots & & & & \\ 0 & \cdots & & I_d & -\lambda I_d \end{bmatrix} + \begin{bmatrix} -\frac{\eta}{\lambda} H_t & \frac{\eta}{\lambda} H_t & 0 & \cdots & 0 \\ 0 & 0 & 0 & \cdots & 0 \\ 0 & 0 & 0 & 0 \cdots & 0 \\ \vdots & & & & \\ 0 & \cdots & & 0 & 0 \end{bmatrix}$$

$$= S_{11} + E.$$

Suppose $|\lambda| \geq 1 - \delta/2$, then $S_{11}$ is invertible. By an identity for the determinant of sum of matrices, we have

$$\det(S_{11} + E) = \det(I + S_{11}^{-1}E)\det(S_{11}),$$

and since $\det(S_{11}) \neq 0$, it must be that $\det(I + S_{11}^{-1}E) = 0$. In other words, $S_{11}^{-1}E$ has an eigenvalue of $-1$. Write

$$S_{11}^{-1} = \begin{bmatrix} D_1 & 0 \\ D_2 & D_3 \end{bmatrix} \in \mathbb{R}^{dh \times dh},$$

where $D_1 \in \mathbb{R}^{d \times d}$, $D_4 \in \mathbb{R}^{d(h-1) \times d(h-1)}$. Then

$$S_{11}^{-1}E = \begin{bmatrix} -\frac{\eta}{\lambda}D_1 H_t & \frac{\eta}{\lambda}D_1 H_t & 0 \\ -\frac{\eta}{\lambda}D_2 H_t & \frac{\eta}{\lambda}D_2 H_t & 0 \end{bmatrix}.$$

By Lemma I.4, we can write

$$S_{11}^{-1}E = F \otimes H_t,$$

where

$$F = \frac{\eta}{\lambda(1 - \delta - \lambda)} \begin{bmatrix} -1 & 1 & 0 \cdots 0 \\ \frac{1}{\lambda} & -\frac{1}{\lambda} & 0 \cdots 0 \\ \vdots & & \\ \frac{1}{\lambda^{d(h-1)}} & -\frac{1}{\lambda^{d(h-1)}} & 0 \cdots 0 \end{bmatrix} \in \mathbb{R}^{d \times d}.$$

The eigenvalues of $S_{11}^{-1}E$ are therefore products of eigenvalues of $F$ and $H_t$. Note that the only nonzero eigenvalue of $F$ is $-\frac{\eta(1+\lambda)}{\lambda^2(1-\delta-\lambda)}$, so there must be an eigenvalue of $H_t$, denoted by $c_i$ such that

$$\frac{c_i\eta(1+\lambda)}{\lambda^2(1-\delta-\lambda)} = 1.$$

Rearranging, and taking the absolute value on both sides, we have

$$c_i\eta|1+\lambda| = |\lambda|^2|1-\delta-\lambda| \geq |\lambda|^2(|\lambda|-(1-\delta)) \geq |\lambda|^2\delta/2.$$

By our choice of $\eta$, $c_i\eta \leq \delta/16$, so we have $\delta(1+|\lambda|) \geq 8|\lambda|^2\delta$, which is a contradiction, because for $|\lambda| \geq 0.5$, $8|\lambda|^2 > (1+|\lambda|)$. We conclude that $|\lambda| < 1-\delta/2$. $\qquad\square$

**Lemma I.4.** *Let $S_{11}$ be the upper left submatrix of dimension $dh \times dh$ in $A_t - \lambda I$, and write $S_{11}^{-1} = \begin{bmatrix} A & 0 \\ C & D \end{bmatrix}$, then $A = \frac{1}{1-\delta-\lambda}I_d$, and*

$$C = \begin{bmatrix} -\frac{1}{(1-\delta-\lambda)\lambda}I_d \\ -\frac{1}{(1-\delta-\lambda)\lambda^2}I_d \\ \vdots \\ -\frac{1}{(1-\delta-\lambda)\lambda^{d(h-1)}}I_d \end{bmatrix}$$

*Proof.* Let $S_{11} = \begin{bmatrix} D_1 & 0 \\ D_2 & D_3 \end{bmatrix}$, where $D_1 \in \mathbb{R}^{d\times d}$ and $D_3 \in \mathbb{R}^{d(h-1)\times d(h-1)}$. Then

$$S_{11}^{-1} = \begin{bmatrix} D_1^{-1} & 0 \\ D_3^{-1}D_2D_1^{-1} & D_3^{-1} \end{bmatrix} = \begin{bmatrix} \frac{1}{1-\delta-\lambda}I & 0 \\ \frac{1}{1-\delta-\lambda}D_3^{-1}D_2 & D_3^{-1} \end{bmatrix}.$$

We prove the lemma by computing the inverse of $D_3$,

$$D_3^{-1} = \begin{bmatrix} -\frac{1}{\lambda}I_d & 0 & 0 & \cdots & 0 \\ -\frac{1}{\lambda^2}I_d & -\frac{1}{\lambda}I_d & 0 & \cdots & 0 \\ -\frac{1}{\lambda^3}I_d & \frac{1}{\lambda^2}I_d & \frac{1}{\lambda}I_d & 0\cdots & 0 \\ \vdots & & & & \\ -\frac{1}{\lambda^{d(h-1)}}I_d & \cdots & & -\frac{1}{\lambda^2}I_d & -\frac{1}{\lambda}I_d \end{bmatrix}$$

We conclude that

$$D_3^{-1}D_2 = \begin{bmatrix} -\frac{1}{\lambda}I_d \\ -\frac{1}{\lambda^2}I_d \\ \vdots \\ -\frac{1}{\lambda^{d(h-1)}I_d} \end{bmatrix}.$$

$\qquad\square$

## J    Additional experimental details

In this section, we report additional experiment details and setups. All experiments are run on a cluster of 4 TPUv3's, and implemented using the Jax and Optax frameworks Bradbury et al. [2018], Babuschkin et al. [2020].

**Linear regression**    We consider two experiments with logistic regression. The first experiment is described in Section 4, and we provide the sweep details below.

| Hypeparameter | Range |
|---|---|
| learning rate | 1e-6, 1e-4, 1e-3, 1e-2, 1e-1 |
| momentum | 0.9, 0.95, 0.99 |
| $\beta_1$ | 0.9, 0.95, 0.99 |
| $\beta_2$ | 0.9, 0.95, 0.99 |
| $\eta_g$ | 1e-10, 1e-5, 1e-3, 1e-2, 1e-1 |

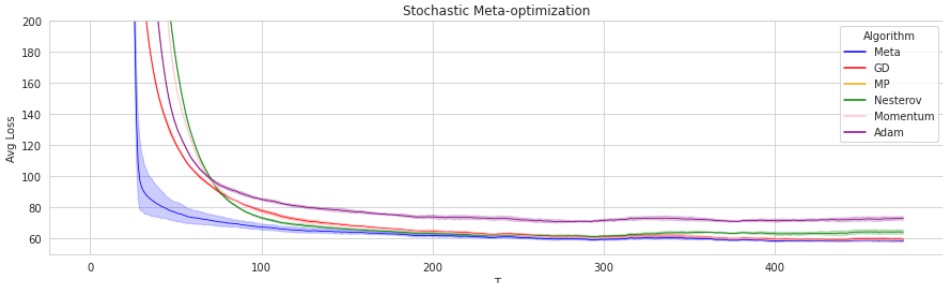

Figure 3: Linear regression with synthetic data. While other baselines have consistent average loss over episodes, indicated by the tight bands, the meta-optimization algorithm is able to learn as more episodes become available. Our algorithm converges faster over time, as demonstrated by the sharp drop in average loss at the beginning of each episode. The Momentum training curve is occluded by the Nesterov momentum curve, and is not as easy to see. The MP algorithm's average loss exceeds 200.

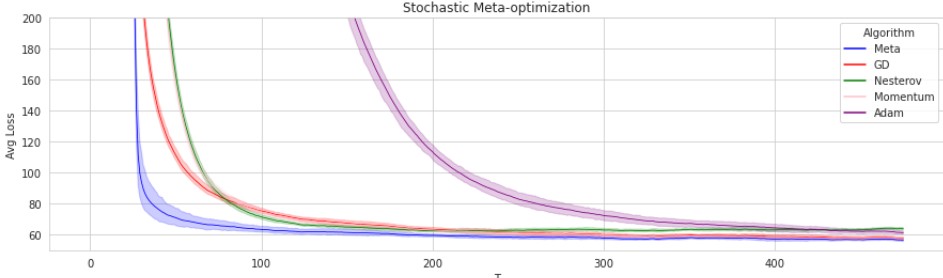

Figure 4: Linear regression with changing synthetic data. Similar to the last experiment, the meta-optimization algorithm is able to learn better updates as more episodes are revealed, while other baseline algorithms do not improve significantly over episodes.

The MP algorithm does not have hyperparameters, but we found that if the gradient updates are not clipped, the algorithm often results in NaN loss values. Therefore, we clip the stepsize of the gradient update to be at most one over the $\ell_2$ norm of the design matrix. For the meta-optimization algorithm, we learn over matrices of size $d \times d$ instead of $3d \times 3d$ as in Algorithm 2, since the disturbances only have $d$ nonzero entries except when reinitializing the iterate. For the base learning rate $\eta$ of the algorithm, we use the best tuned learning rate for GD, which is 1e-3, and only tune $\eta_g$.

In addition to GD, MP, Nesterov momentum, and Momentum, we also compare with Adam, and plot the overlayed episodic loss in Figure 3. The bands around the training curves indicate the range of losses over the $N$ episodes, and the solid line is the mean.

In the second experiment, we add episodic noise to the design matrix. The experiment setup is the same as in the first experiment, but in each episode, we add an additional entry-wise Gaussian noise with standard deviation 0.5 to the design matrix. Compared to the first experiment, the objective functions have larger shift between time steps and episodes, and thus the task is more difficult. We tune the algorithms using the same range of hyperparameters, and plot the moving average of their losses in Figure 4. We do not compare with the MP algorithm in this experiment, since it requires parameters that depend on the Hessian of the objective functions, which is changing in this setting.

**Logistic regression for MNIST classification** We consider learning a multinomial logistic regression model for the task of MNIST classification. When preprocessing data, we split the training dataset into 50k training examples and 10k validation examples. We again consider the episodic setting; we run for 5 episodes and in each episode, we train for 50 epochs on the training examples with a batch size of 256. Between episodes, we shuffle the training examples and reset the model parameters to the same initialization, randomly generated from a Gaussian distribution. We compare

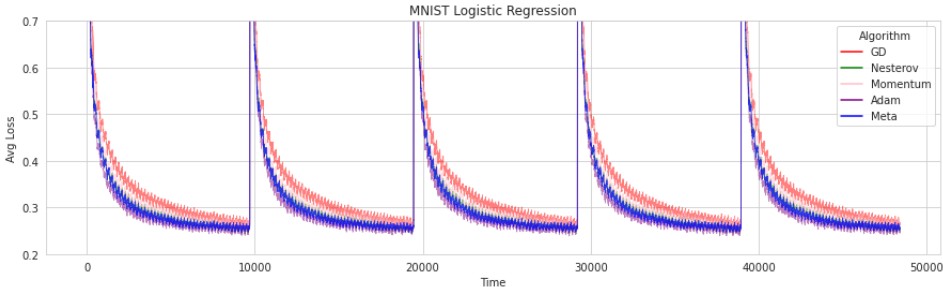

Figure 5: Logistic regression for MNIST classification. Meta-optimization outperforms SGD and momentum. Nesterov momentum is on par with meta-optimization (line occluded in the figure), and Adam performs the best out of all the algorithms. Adam's performance is possibly due to the inclusion of $\beta_2$. Note that even though in theory meta-optimization can compete with a single fixed preconditioner, similar to the guarantee of AdaGrad, Adam uses a moving average of the second moment instead of the diagonal AdaGrad update.

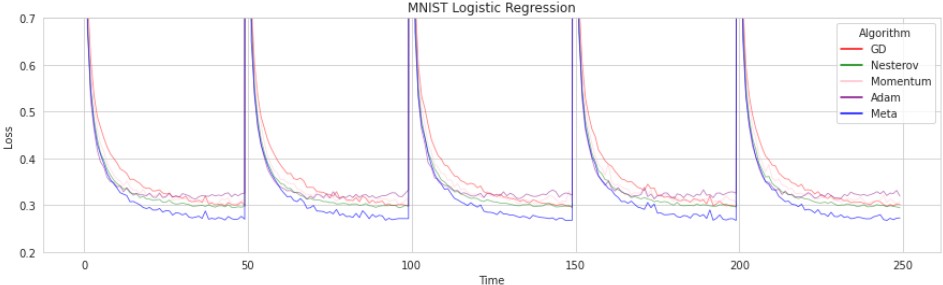

Figure 6: Validation losses on MNIST with logistic regression. Meta-optimization does surprisingly well in generalization, outperforming all baselines including Adam.

with SGD, Nesterov acceleration, Momentum, and Adam, and plot the moving average of their training losses in Figure 5 with a window of 100 steps. In addition, we evaluate the loss on the validation examples every epoch, and plot the results in Figure 6. We use the Optax implementations for the baseline algorithms, and give their tuning details in the table below.

| Hypeparameter | Range |
|---|---|
| learning rate | 1e-5, 1e-4, 1e-3, 1e-2, 1e-1, 1.0, 10. |
| momentum | 0.9, 0.95, 0.99 |
| $\beta_1$ | 0.9, 0.95, 0.99 |
| $\beta_2$ | 0.9, 0.95, 0.99 |
| $\delta$ | 1e-5, 1e-4, 1e-3, 1e-2 |
| $\eta_g$ | 1e-10, 1e-5, 1e-3, 1e-2, 1e-1 |

For our meta-optimization algorithm, we use the best learning rate from SGD as $\eta$, and tune $\eta_g$, the meta learning rate, and $\delta$, the regularization parameter. The final values are: $\delta = 0.0001$, $L = 3$, $\eta_g = 0.01$, $\eta = 1.0$. Since this setting is relatively high-dimensional, instead of learning over matrices as in Algorithm 2, we learn over scalar-valued $M^l$'s. The algorithm is still well-defined, and this variant is more practical for large-scale problems.

**Neural networks for MNIST classification** We consider two experiments for classification on MNIST with neural networks. The first experiment is given in the main text, where we investigate the behavior of meta-optimization given a suboptimal learning rate. In the following experiment, we perform a sanity check and show that there is also improvement when the learning rate is set to the optimal 1.0. The architecture of the network and the hyperparameter sweep ranges for both experiments are the same, and the sweep ranges are as the table above. The plot demonstrates that

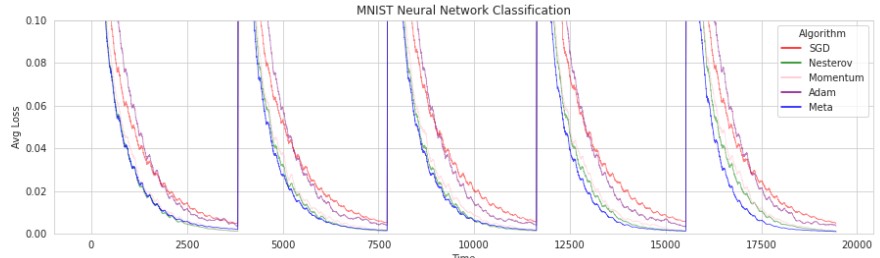

Figure 7: Comparison plot for the training performance of meta-optimization and other baselines on the MNSIT classification task.

given this learning rate, meta-optimization can improve upon not just SGD, but also Adam and momentum.