# OpenReview forum: "Online Control for Meta-optimization"
_NeurIPS.cc/2023/Conference — NeurIPS 2023 spotlight_

### Official Review · Reviewer_MnL8 · 2023-06-27

**Soundness:** 4 excellent
**Presentation:** 2 fair
**Contribution:** 3 good
**Rating:** 7
**Confidence:** 4

**Summary:**

This paper studied the problem of meta-optimization through the perspectives of online control. In this paper, the meta-optimization problem is thought as a sequence of episodic optimization problem and the performance is measured by the regret against the best static optimizer belonging to a convex class. By re-writing the gradient descent update rule into a LTV system with non-stochastic disturbance, the author presented a control-motivated meta-optimization algorithm can achieve nearly-optimal optimization performance in hindsight. The main theorem shows that the algorithm proposed by this paper can achieve sub-linear regret for both quadratic and convex, smooth costs.

**Strengths:**

I think the approach taken by this paper is quite clever in that it re-write the extremely difficult, highly non-convex meta-optimization problem into a much more tractable online learning problem. The idea seems quite general and I encourage the authors to extend this work to meta-optimize more sophisticated algorithms beyond standard gradient descent (e.g. momentum or Adam).

The use of regret to benchmark the performance of meta-optimization also seems to be a good idea, as it balances the desire for fast convergence and quality for the final solutions.

Given this is a ML venue, this paper did a good job with introducing the relevant notations and terminologies from control.

**Weaknesses:**

While I think this paper has a lot of potential, I regrettably cannot given a strong rating because I feel the presentation is overall quite lacking.

1. I do not agree with the author's claim that their setting of meta-optimization generalizes hyper-parameter optimization. While, the algorithm in this paper can yield comparable performance to the best hyper-parameters in hindsight, I struggle to find how a good set of hyper-parameters can be extracted this algorithm.

2. This paper seems a bit carried away with the control perspective and fails to give concrete examples how the results can be applied to practical learning problems. Although an example is given in Appendix I, it is rather simplistic. And the lack of source code makes it hard for the readers to understand how the algorithms are implemented.

3. Some of the notations are not very well-specified. a) the disturbance $w_{t, i}$ is not explicitly defined, is it $\nabla f_{t, i}(x_{t-1} i)$ as given in eq. (3)? b) in Theorem 2, what is the function class $\Pi$, is it related to $\Pi_{DFC}$?

4. Regarding the equation between lines 196 and 197, isn't it a second order approximation of gradient descent? This simplifying assumption should be discussed more thoroughly.

5. Algorithm 2 should be more self-contained, i.e., eq. (7) should be in the body. Also I would love to see more description on this algorithm.

6. Perhaps my greatest complaint. The experiments are very simple and does not reflect any realistic scenarios where meta-optimization are helpful. Linear regression and MNIST are both too small that vanilla SGD can usually get very good performance already. Also, I very much don't like the decision to move almost all of the experimental details into the appendix.

**Questions:**

See my concerns above. I am open to raising my rating significantly if author can either prove me wrong or offer fixes to these issues.

**Limitations:**

This paper is mostly theoretical and addresses fundamental problems in optimization for ML. So, I do not envision any concerns over ethics.

However, the authors absolutely should include source code for their experiments, especially since their proposed algorithm is quite involved.

---

> ### Author Rebuttal · Authors · 2023-08-07
>
> We thank the reviewer for taking the time to review our work and the valuable feedback. We first address the point of simple experiments, and then address other comments one by one below.
>
> **please also see pdf for all reviewers, as it contains experiments**
>
> We would like to emphasize that this is mainly a theoretical paper, and in our opinion this work has sufficient novelty to stand alone as a conceptual and theoretical contribution. To elaborate on the difficulty of getting this result, and the techniques from different subfields that we needed to incorporate to circumvent known NP-hardness results:  we change the formulation of the problem and incorporate convex relaxation (point 1 below), we give a new dynamical system formulation of optimization (which is different from all previous formulations), and we use new techniques from online control to solve this dynamical systems formulation (which have not been used in optimization before). Notice our method can also compete with gradient descent with momentum, and preconditioned methods, which is a strength you wanted to see in future work (and is already accomplished in this paper).
>
> To support the theory, we provide proof-of-concept experiments, rather than extensive evaluations as in applied papers. The existing experiments are consistent with the convex regime studied in the paper, and we include an additional experiment on neural networks in the general response. Even though SGD can solve these problems, it still requires hyperparameter tuning to achieve the best performance, and as shown experimentally, meta-optimization has improved performance over time. Extension to non-convex meta-optimization and developing practical algorithms suitable for ultra large-scale optimization is important future work. These experimental results also demonstrate that there is potential for this new approach to be applied to a broader set of problems, including nonconvex optimization with large-scale deep neural networks.
>
> We respond to your other comments below:
>
> 1. Directly optimizing over the hyperparameters is a non-convex problem in general, and non-convex problems are NP-hard in the worst case. We use an improper learning technique, meaning that we do not directly manipulate the hyperparameters but the optimization iterates themselves. This improper learning technique is the result of a convex relaxation, and this combination of techniques to tackle non-convex problems is commonly used in the theoretical machine learning community. A prominent example is the LASSO algorithm: it relaxes the sparsity constraint to an $\ell_1$ constraint. This circumvents the NP-hard problem of recovering the sparse coefficients, and allows for equal expressive power in prediction/classification. (Note there is a large literature on statistical assumptions that allow for recovery of the sparse coefficients, we do not refer to these). Other examples include: online nonstochastic control (which we make use of), trace norm relaxation for matrix completion, and many more.
>
>     The notion of competing with the best-in-hindsight in terms of performance, a.k.a. regret minimization, is also a widely adopted paradigm. Notably, adaptive gradient methods (starting from AdaGrad) were developed in this context.
>
>     For hyperparameter optimization, our approach which makes use of convex relaxation and regret minimization, improves upon grid search of hyperparameters in terms of the iterates. In particular, if we can compete with the best optimizer in hindsight, the optimization iterates will converge to the optimum faster over many episodes, compared to grid search.
>
>     Lastly, the relationship between the best disturbance-response controller and the optimal state feedback controller, which directly encodes the hyperparameters, is an interesting research topic. Indeed, [1] shows that in the LQR setting under certain conditions, parameters of the disturbance-response controllers are good approximations of the optimal feedback controller.
> 2. We describe the control approach in detail because it is the foundation for our algorithm. The example in Appendix I is simple to illustrate the theoretical approach. In appendix H, we give examples of optimization algorithms that we can compete against, which include essentially gradient descent with momentum, and preconditioning methods. We plan to open source our code in the coming weeks and make our method more widely available as soon as we can.
> 3. Thank you for pointing them out, we will clarify in the revision. (a) Yes, $w_{t, i}$ is as defined in (3) during an episode, and as in (4) at the end of an episode. (b) The algorithm class $\Pi$ and its relationship to $\Pi_{DFC}$ is explained in section 3.2.
> 4. This equation represents the relationship between the previous gradient and the current gradient. In the convex quadratic case, the $H_{t, i}$ matrix is the Hessian; in the convex smooth case, there always exists such a linear transformation and we consider a particular one, as stated in the appendix. The linear transformation is given by second-order information of the function at different points between the previous and current iterate.
> 5. Thank you for the suggestion, we will include the ideal cost and more exposition on the algorithm in the main body. We have also commented on the intuition behind the ideal cost in the response to reviewer yXWX.
>
> [1] On the Relationship of Optimal State Feedback and Disturbance Response Controllers
>
> We hope our response answers your questions and that you consider raising your score.

---

> > ### Comment · Reviewer_MnL8 · 2023-08-10
> >
> > Thank you for your response. I am happy that you have addressed much of my concerns with the motivation of this paper. Originally, I thought your goal is to do hyper-parameter tuning, but now I understand that this paper is about the application of online learning to a more general notion of meta-optimization.
> >
> > Regarding with your responses:
> > 1. I am aware of the literature you pointed out. I think a better angle of justifying your algorithm is by considering problems with time-varying optimization objective. Please correct me if you think my intuition is wrong -- I think the proposed algorithm is a very neat way to "adapt" to a sequence of evolving learning tasks given to the learner, and the regret guarantee would be highly relevant in such a scenario. I think that the emphasis on hyper-parameter tuning may mislead the readers.
> >
> > 2. I am not happy with this response. From my understanding, you directly transplanted online learning/control methods to solve a learning problem. But this paper does not justify why and how such formulation is applicable or natural for a learning problem and explain that the terminologies are not artifact of limitations in control. In fact, I am now convinced that this work is very strong from a theoretical perspective, but the motivations are still somewhat weak. The answer I am looking for here is more discussions why this paper's formulation is not overfitting to existing methods in online learning/control.
> >
> > 3. a) thanks, b) can you just directly tell me what is the compactor class $\Pi$ used in Theorem 3.2? I still cannot find it.
> >
> > 4. I see. But I am now a little confused by how does your algorithm compute the $A$ matrices when the cost is a general smooth function.
> >
> > 5. I like your explanation of the ideal cost. It would be lovely to see this discussion being added to the main body.
> >
> > Lastly, the experiments are still kind of simple, but I appreciate the efforts into addressing my concerns.
> >
> > Overall, I am convinced of this paper's technical contributions. Nevertheless, I still feel that the discussions leading up to the main results could be made more clear. I will raise my score to **5** and would be inclined to make further changes if my remaining concerns could be answered this week.

---

> > > ### Author Response · Authors · 2023-08-11
> > >
> > > Thank you for your fast response, engagement, and openness. We address your comments point by point below.
> > >
> > > 1. We chose hyperparameter optimization as an example that is important and everyone is familiar with, but we agree that we consider a much more general setting. We will emphasize that in the final version.
> > >
> > > 2. We are happy to elaborate on this issue, since indeed, this touches upon the very heart of our approach and problem we tackle. The history of control methods in mathematical optimization is vast, and we give a survey in Appendix A and B. In fact, some of the most well known methods in optimization are directly motivated by intuition from dynamical systems; for example, the Polyak “heavy ball” method, which inspired momentum, is influenced by Newtonian physics as its very name suggests.
> > > The standard methodology that connects optimization and control is Lyapunov’s stability theory. Deriving the convergence of various optimization algorithms using Lyapunov stability analysis is surveyed in Wilson’s recent thesis [1]. However, this approach is insufficient for our purpose, because of two reasons:
> > >     - The problem we address, of competing with the best algorithm, is nonconvex. Thus, stability arguments would imply convergence to a local optimum, which would not give the desired outcome. That is why we resort to a different control formulation stemming from optimal control, and the use of new techniques in online control based on convex relaxation. These techniques guarantee competitiveness with the global optimum (as we explained in the LASSO example for the previous point).
> > >     - Lyapunov’s stability analysis can show the stability of a given system (correspondingly the convergence of a given optimization algorithm). However, our goal is to design meta-optimization algorithms that can compete with the best method in hindsight, which goes beyond the scope of analyzing convergence of existing algorithms.
> > >
> > >     We will clarify these motivations in the revision.
> > >
> > > 3. b) Yes of course. The characterization is the equation below Line 290: basically the policy class amounts to linear functions of past gradients.  As explained in Lines 291 to 294, for deterministic meta-optimization, it captures gradient descent, momentum, and preconditioning methods on pseudo-gradients.
> > >
> > > 4. This is an excellent question. The matrices A exist, as we prove, but we do not need to compute them. The algorithm applies a linear policy to the disturbances, which we can compute (because the disturbances are given by the gradients), and does not make use of A. The existence of a linear dynamical formulation is important for the analysis, but not for the algorithm. Not having access to A implies that we cannot obtain the full gradients for updating the M matrices, but we can estimate these gradients using zero-order methods, as we detail in Appendix E.
> > >
> > > 5. We will be happy to add this!
> > >
> > > [1] Lyapunov Arguments in Optimization. Wilson, Ashia, 2018.
> > >
> > > We hope this answers your questions and you are open to further raising the score.

---

> > > > ### Comment · Reviewer_MnL8 · 2023-08-11
> > > >
> > > > Thank you for the detailed response that addressed all of my concerns, and I am looking forward to reading the final version. I will further raise the score to **7**.

---

> > > > > ### Comment · Reviewer_MnL8 · 2023-08-11
> > > > >
> > > > > BTW, I have another comment. I feel that some important information are buried in the Appendix and not referenced in the main body. I suggest the authors to keep this in mind in their revision, as I wasn't able to catch some of the key messages in the Appendix during my initial read-through.

---

> > > > > > ### Author Response · Authors · 2023-08-11
> > > > > >
> > > > > > Thank you for pointing this out, we will attempt to incorporate more pointers to the appendix in the revision.

---

### Official Review · Reviewer_fcBo · 2023-07-03

**Soundness:** 3 good
**Presentation:** 3 good
**Contribution:** 3 good
**Rating:** 7
**Confidence:** 3

**Summary:**

This paper proposes a framework for optimization whose goal is to learn the best optimization algorithm from experience, and gives an algorithmic methodology using feedback control for this meta-optimization problem. The authors derive new efficient algorithms for meta-optimization using recently proposed control methods, and prove sublinear meta-regret bounds for quadratic and convex smooth losses. The approach leverages convex relaxation techniques in the recently-proposed nonstochastic control framework to overcome the challenge of nonconvexity. Consequently, it enables us to learn a method that attains convergence comparable to that of the best optimization method in hindsight from a class of methods.


**Strengths:**

This paper introduces a novel approach based on control theory for the task of meta-optimization – online learning of the best optimization algorithm. It provides a novel control formulation for meta-optimization based on the recently proposed framework of online nonstochastic control. Consequently, a new metric for meta optimization called meta-regret is proposed, which measures the total cost compared to the cost of the best algorithm in hindsight in meta-optimization. FurthermoreMoreover, this is a theoretical paper of high technical quality. It derives efficient algorithms for meta-optimization based on recently proposed Gradient Perturbation Controller (GPC), a new type of online feedback-loop controller. Furthermore, this paper provides the analysis and proof of the tight sublinear regret bounds for quadratic and convex smooth objective functions.


**Weaknesses:**

This paper is well organized and clearly written in general. However, in algorithm 2 (Line 9), Eq. (7) is missing in the main paper, although we may find this Eq. in Appendix. This is mainly a theory paper, in section 4, it provides an proof of concept on experiments with quadratic and convex smooth losses. We are curious to investigate the empirical impact of meta-optimization on board applications. Therefore, it might strengthen the impact if the authors could provide the experiment on non-convex loss, for example, a classification task using simple neural network.


**Questions:**

This is a very solid theory paper. However, in the main paper, the experiment part is limited. Can we move Logistic regression for MNIST classification experiment from Appendix and add to main paper?  In Figure 1, for y-axis, is it the moving average of objective values on training data or validation data?


**Limitations:**

There is no code provided to reproduce the results.

---

> ### Author Rebuttal · Authors · 2023-08-07
>
> We thank the reviewer for taking the time to review our work and their valuable feedback. We are happy to address the weaknesses:
> 1. We will provide explanations and intuitions for the ideal cost in the main paper, for clarity and completeness.
> 2. We have an additional experiment on MNIST classification with a neural network. Comments and details on the experiment are in the top-level response.
> 3. We definitely plan to open source our implementation in the coming weeks, as soon as we can.
>
> For the questions:
> 1. We can certainly move the logistic regression experiment to the main paper.
> 2. In Figure 1, the objective values are on the training data, since we study optimization instead of generalization.

---

> > ### Comment · Reviewer_fcBo · 2023-08-15
> > **Thanks to the authors for the rebuttal**
> >
> > I’ve read comments from all the other reviewers. Thank you for your rebuttal, and I appreciate that my concerns have been addressed.

---

### Official Review · Reviewer_PoEe · 2023-07-04

**Soundness:** 4 excellent
**Presentation:** 3 good
**Contribution:** 4 excellent
**Rating:** 7
**Confidence:** 2

**Summary:**

This paper proposes an online control method to meta optimization problem. The authors formulate the problem into a robust control problem and leverages non-stochastic control framework to achieve convex relaxation. The regret guarantees are derived theoretically and experiments show its superiority compared with the best optimizers in hindsight.

**Strengths:**

The main text is very concise in presenting the main technical discovery of the paper, as well as necessary background discussion and experiment results. The appendix provides very comprehensive introduction of technical background, proof of theorems and more interesting experiment results. This paper is overall well written and solid in theory.

**Weaknesses:**

I personally think the experiment presented in the main paper can be replaced by a more complex one. For example, could you consider experiments with neural network training using Adams, which will be more interesting to modern AI field. Or any other complex experiments.

**Questions:**

In line 7 of Algorithm 2, how is the gradient information calculated. If it is calculated through approximation (e.g., finite-difference), how will the approximation error affect the final outcomes?

Is the meta regret of the proposed method always positive? Because intuitively, the best optimizer in hindsight should always perform better. If yes, why the proposed method can achieve lowest loss in all experiments compared with all other optimizers?

**Limitations:**

The limitations of the proposed method is not explicitly discussed in conclusion, although future directions are pointed out.

---

> ### Author Rebuttal · Authors · 2023-08-07
>
> We thank the reviewer for taking the time to review our work and the valuable feedback. Given your suggestion, we have added a proof-of-concept experiment for neural network classification on the MNIST dataset, with Adam as one of the baselines. Additional comments and details on the experiment are in the top-level response.
>
> To address your questions:
> 1. In Line 7 of Algorithm 2, we assume access to the exact gradients of the function. If we use finite-difference, under Lipschitz loss functions, the approximation error of the gradients will be added linearly to the regret; in other words, the sum of all biases will be added to the final regret. If the gradients are stochastic with zero mean noise, the expected regret will remain the same.
>
> 2. The meta-regret of a method can be negative if the method does better than the best optimizer in the benchmark algorithm class $\Pi$. The proposed method can compete with a class of optimizers that include preconditioned methods (see appendix G and H), and Newton’s method converges in one step for quadratic minimization problems. Our baselines in the experiments are mostly first-order, since they are the most popular optimization algorithms.
>
> Given the positive evaluation of novelty and contribution in this review, and the additional nonconvex optimization experiment, we hope the reviewer is open to raising their score.

---

> > ### Comment · Reviewer_PoEe · 2023-08-15
> > **Thank the authors for the response**
> >
> > I have read the authors' responses and other reviewers' comments. I think the authors satisfactorily addressed my concerns. I am glad to raise my score.

---

### Official Review · Reviewer_yXWX · 2023-07-07

**Soundness:** 4 excellent
**Presentation:** 4 excellent
**Contribution:** 4 excellent
**Rating:** 8
**Confidence:** 3

**Summary:**

The paper considers a new framework, meta-optimization. To solve this problem, the authors propose an online control formulation with linear time-varying dynamics. Moreover, a novel algorithm is proposed to solve meta-optimization and shown to enjoy sublinear regret under both the quadratic loss and convex smooth losses. As an application, one can formulate the hyperparameters, such as learning rate, as a meta-optimization problem. Finally, the authors provided experiments to show the algorithm's superior performance compared to existing hyperparameter tuning benchmarks.

**Strengths:**

1. The authors consider a new framework, meta-optimization, to choose the optimal hyperparameters.
2. The authors point out that meta-optimization with gradient descent methods can be viewed as an online control formulation with linear dynamics.
3. A new algorithm is proposed to solve the meta-optimization problems inspired by the online nonstochastic control formulation. New techniques are used to show the sublinear regret for meta-optimization with quadratic and convex smooth losses.
4. Comprehensive experiments are conducted and comparisons to benchmarks are provided.
5. The paper is well-organized and well-written.

**Weaknesses:**

1. Several online optimization problems are proposed, while not all of them are used to reach the final goal. The authors may reorganize the presentation and clarify the relation between the formulations.
2. It is preferred if the authors can mention the techniques used in the main text following the main theorems.

**Questions:**

I have some minor questions:
1. The formulation proposed in Lines 57-58 seems not to be used in the analysis. Equation (5) is a more concrete formulation for meta-optimization. Could you explain why to need this generic formulation and how it is related to other formulations?
2. The online control formulation (1) - (3) involves a control variable $u_{t, i}$ while it is missing in (5). What is the meaning of this control signal for meta-optimization and, more concretely, in learning gradient descent step size?
3. What is the intuition behind using ideal cost $g_{t, i}$ in Algorithm 2?

**Limitations:**

The limitations and future work have been addressed in the conclusion section.

---

> ### Author Rebuttal · Authors · 2023-08-07
>
> We thank the reviewer for the positive evaluation of our work and the helpful suggestions. To address the weaknesses, we are very happy to clarify the online optimization formulations in the revision, and include more techniques that are used in proving the main theorems.
>
> For the questions:
> 1. This formulation uses short-hand notation on the episodic loss, but it is not necessary. It is the same as (5), which expands the episodic loss. We can remove the redundant one for the final version.
> 2. The control variable in (5) is implicit in $x_{t, i}^\pi$, where $\pi$ denotes the control policy. $x_{t, i}^\pi$ is the state reached under the policy $\pi$ at time $(t, i)$, so while the control cost $f$ doesn’t take the control signal $u_{t, i}$ into account, the state is still affected by the control policy. In meta-optimization, the control signal can be used to simulate the update of any optimization algorithm; therefore, through learning the best controls, we can compete with the best optimizer. The control signal, including learning the gradient descent step size, is a linear function of past disturbances, which largely depends on the past gradients.
> 3. We use the ideal cost because we need a loss function that depends only on $M$ to update it. The original control cost is a function of $x_{t, i}$, which depends on $M_{t, i}, M_{t-1, i}, \cdots$. On the other hand, the ideal cost is the cost that we would have incurred, if we started from the zero state and executed $M_{t, i}$ for $L$ time steps. It is a function of only $M_{t, i}$. There are two reasons why using the ideal cost works for non-stochastic control, which we elaborate in the appendix. In summary, the reasons are:
> - The system is stable, and therefore the control cost depends minimally on the controllers executed before $L$ time steps, e.g. $M_{t-L, i}, M_{t-L-1, i}, \ldots$.
> - We update the $M_{t, i}$’s with a memory-based OGD algorithm developed under the online convex optimization with memory framework, which accounts for the fact that the original control cost can depend on $L$ inputs, and still gives us the desired guarantee.

---

> > ### Comment · Reviewer_yXWX · 2023-08-17
> >
> > Thanks for your response. My concerns have been well-addressed. This is an excellent theoretical work.

---

### Author Rebuttal · Authors · 2023-08-07

**please see attached pdf**

We thank all reviewers for their time and valuable feedback. One common suggestion is including more complex experiments, for example with neural networks. Since the main contribution of this paper is theoretical, the experiments are proofs-of-concept to show the potential of this theoretical framework. The existing experiments are in the convex regime, consistent with theory and empirically support our theoretical results.
For the purpose of showing potential of future applications/theory to practical optimization settings, we give a proof-of-concept additional experiment in the non-convex regime, namely MNIST classification with a neural network in the uploaded PDF file.

As can be seen in the first experiment, given a good base learning rate ($\eta$ in the algorithm), meta-optimization improves over SGD, Adam, and momentum (with tuned hyperparameters). The performance gap also widens as there are more and more episodes. In the second experiment, we give meta-optimization a suboptimal base learning rate (0.5), and show that the meta optimizer not only outperforms SGD with lr=0.5, but also SGD with lr=1.0, and approaches the performance of momentum, the best baseline. Given these initial positive results, we believe that the meta-optimization framework holds much promise and this work lays a solid theoretical foundation for future research.

Experimental details: We consider the MNIST classification task with a neural network. The network has 3 layers and is fully connected, the layer sizes are: [784, 512, 10]. We run for 5 episodes with 20 epochs in each episode. Since this problem is high-dimensional, we choose to learn scalar weights of past gradients. We take L=20, the meta learning rate = 0.005, and we have found that the results are insensitive to the meta-learning rate choice. The network is trained using stochastic gradients from batches of size 256. The hyperparameters of the baselines are tuned, and the search space is the following:

SGD: learning rate = [1e-5 to 10] with increments of factor 10

Momentum: learning rate = [1e-5 to 10] with increments of factor 10, momentum = [0.9, 0.95, 0.99]

Nesterov momentum: learning rate = [1e-5 to 10] with increments of factor 10, momentum = [0.9, 0.95, 0.99]

Adam: learning rate = [1e-4, 1e-3, 1e-2], beta_1 = [0.9, 0.95, 0.99], beta_2 = [0.9, 0.95, 0.99]
For Adam, the learning rates 1e-4 and 1e-2 were already suboptimal, so we did not search beyond them.

---

### Decision · Program_Chairs · 2023-09-21

**Decision:**

Accept (spotlight)

**Comment:**

This paper casts meta-optimization (the task of learning the best optimization algorithm) in terms of control theory, and uses convex relaxation techniques to obtain regret guarantees, despite the non-convexity of the problem.

The reviewers unanimously agree that the paper is well-written, and its contributions merit acceptance. I agree with them and think that this paper will spark fruitful discussions at the intersection of optimal control and meta-optimization. Consequently, I recommend this paper be accepted as a spotlight.

Please incorporate the reviewer's comments in the final version of the paper. In particular, addressing the following concerns will help strengthen the current version of the paper:
- Add the non-convex MNIST experiments that were included as part of the rebuttal, and open-source code.
- Provide better intuition and exposition behind the ideal cost and resulting algorithm.